# RoVer: Robot Reward Model as Test-Time Verifier for Vision-Language-Action Model

## Abstract

Vision-Language-Action (VLA) models have become a prominent paradigm for embodied intelligence, yet further performance improvements typically rely on scaling up training data and model size – an approach that is prohibitively expensive for robotics and fundamentally limited by data collection costs. We address this limitation with **RoVer**, an embodied test-time scaling framework that uses a **Ro**bot Process Reward Model (PRM) as a Test-Time **Ver**ifier to enhance the capabilities of existing VLA models without modifying their architectures or weights. Specifically, RoVer (i) assigns scalar-based process rewards to evaluate the reliability of candidate actions, and (ii) predicts an action-space direction for candidate expansion/refinement. During inference, RoVer generates multiple candidate actions concurrently from the base policy, expands them along PRM-predicted directions, and then scores all candidates with PRM to select the optimal action for execution. Notably, by caching shared perception features, it can amortize perception cost and evaluate more candidates under the same test-time computational budget. Essentially, our approach effectively transforms available computing resources into better action decision-making, realizing the benefits of test-time scaling without extra training overhead. Extensive experiments demonstrate that RoVer consistently improves success rates across diverse manipulation tasks. Our contributions are threefold: (1) a general, plug-and-play test-time scaling framework for VLAs; (2) a PRM that jointly provides scalar process rewards and an action-space direction to guide exploration; and (3) an efficient direction-guided sampling strategy that leverages a shared perception cache to enable scalable candidate generation and selection during inference.

## 1 Introduction

The field of embodied AI has witnessed significant progress, from specialist models such as ACT, DP, and DP3 (Zhao et al., 2023; Chi et al., 2024; 2023; Ze et al., 2024) to general-purpose VLA systems like OpenVLA, $\pi 0$, and RDT (Kim et al., 2024; Black et al., 2024; Liu et al., 2025c). These VLA models are typically trained end-to-end on paired vision–language–action data, and most gains to date have come from training-time scaling strategies, i.e., using larger backbones, diversifying demonstration data, and enhancing data curation processes. However, even under identical settings, VLA success rates often fluctuate due to stochastic decoding and manipulation brittleness, and the issue becomes particularly pronounced on long-horizon tasks. This phenomenon motivates reallocating part of the effort from training-time scaling to inference-time computation, with the goal of stabilizing outcomes and unlocking the latent capabilities of existing VLA models.

Inspired by the tremendous success of scaling laws (Kaplan et al., 2020) in large language models (LLMs) and vision-language models (VLMs), many research efforts in embodied AI have aimed to replicate this progress by collecting large-scale training datasets, such as Open-X-Embodiment, Agi World, Fourier ActionNet, and ARIO (Collaboration et al., 2023; AgiBot-World-Contributors et al., 2025; Fourier ActionNet Team, 2025; Wang et al., 2024), and increasing model parameter sizes (e.g., RT-X (Collaboration et al., 2023), OpenVLA (Kim et al., 2024), RDT (Liu et al., 2025c)). However, unlike internet-scale corpora for LLMs/VLMs, the acquisition of diverse, high-quality robotic experience data remains a resource-intensive and time-consuming process, creating a significant bottleneck for training-time scaling in embodied AI field.

To enhance VLA performance without further training, researchers have begun to increase the proportion of 'thinking' at test-time. Some embodied approaches adopt a fast-slow-thinking design that separates a deliberative high-level 'brain' from a low-latency 'cerebellum' controller, enabling planning-execution decoupling for more challenging tasks(Kahneman, 2011; Zhang et al., 2025; Shentu et al., 2025; Bu et al., 2025a) . Subsequent works expand supervisory signals by injecting chain-of-thought annotations into datasets to train sophisticated high-level reasoning modules that can be invoked during inference(Zawalski et al., 2025; Ji et al., 2025; Zhou et al., 2025). These avenues have already shown their potential in LLMs and VLMs, often referred to as test-time scaling(Liu et al., 2025b; Ma et al., 2025). However, most of the above methods require expanding the original dataset with supplementary annotations.

In this case, we pose a fundamental question: can we leverage test-time scaling mechanisms to enhance VLA performance *without* requiring additional data or model retraining? This question leads us to explore a promising paradigm that focuses on maximizing the utility of existing VLA models during inference, rather than pursuing increasingly larger datasets and architectures. To this end, we introduce **RoVer**, a novel external process reward model designed to enhance the capabilities of frozen VLA policies purely at inference time. In particular, our RoVer not only evaluates the reliability scores of massive candidate actions, but also forecasts precise 6D refinement directions in the action space. This design enables both intelligent candidate evaluation and direction-guided sampling, all while maintaining the original backbone architecture and weights intact. Crucially, our approach transforms additional test-time computational resources into improved decision-making capabilities, effectively circumventing the traditional constraints of data collection and model retraining. Another elegance of RoVer lies in its ability to leverage shared computation efficiently. By caching and reusing intermediate features from the perception pipeline, our framework can evaluate multiple action candidates with minimal additional computational overhead. This design choice ensures that the increased decision-making capability comes at a reasonable computational cost, making it practical for real-world robotic applications. In summary, our work makes three significant contributions to the field:

- A general, plug-and-play test-time scaling framework that enhances frozen VLA policies purely at inference time, demonstrating a new paradigm for improving VLA performance without the traditional costs of data collection and model retraining.

- A compact yet powerful Process Reward Model that simultaneously generates scalar process rewards and refinement directions of candidate actions, enabling effective verifier-guided exploration in the action space.

- An efficient direction-guided sampling strategy that capitalizes on a shared *perception cache*, effectively amortizing perception costs and enabling the evaluation of more candidates within the same computational budget.

## 2 RELATED WORK

**Vision-Language-Action Models.** VLA models have evolved from specialized, task-specific policy Zhao et al. (2023); Chi et al. (2024; 2023); Ze et al. (2024) to general-purpose systems that couple perception, language, and control Kim et al. (2024); Black et al. (2024); Liu et al. (2025c), with recent advancements exploring hierarchical architectures inspired by human cognition's dual-process theory Kahneman (2011); Shentu et al. (2025); Zhang et al. (2025); Bu et al. (2025a); Cui et al. (2025); Liu et al. (2025a); NVIDIA et al. (2025). Most performance gains to date have come from *training-time scaling* —larger backbones, broader demonstrations, and improved data curation.

**Test-time Scaling in LLMs.** Recent advancements in large language models (LLMs) have highlighted the efficacy of test-time scaling (TTS) techniques in boosting their reasoning capabilities during the inference phase. TTS approaches can generally be categorized into two main types(Liu et al., 2025b): internal and external. Internal TTS methods, such as self-reflection and Chain-of-Thought (CoT), involve the model generating and refining its own intermediate reasoning steps, exemplified by models like OpenAI o1 and DeepSeek-R1(OpenAI et al., 2024; DeepSeek-AI, 2025). In contrast, external TTS leverages a Process Reward Model (PRM) to provide feedback and supervise the generation process, facilitating search strategies like Best-of-N (BoN) and Beam Search. Notably, studies have indicated that with the aid of external TTS, even significantly smaller models can, in certain tasks, achieve or surpass the performance of much larger counterparts. This paradigm

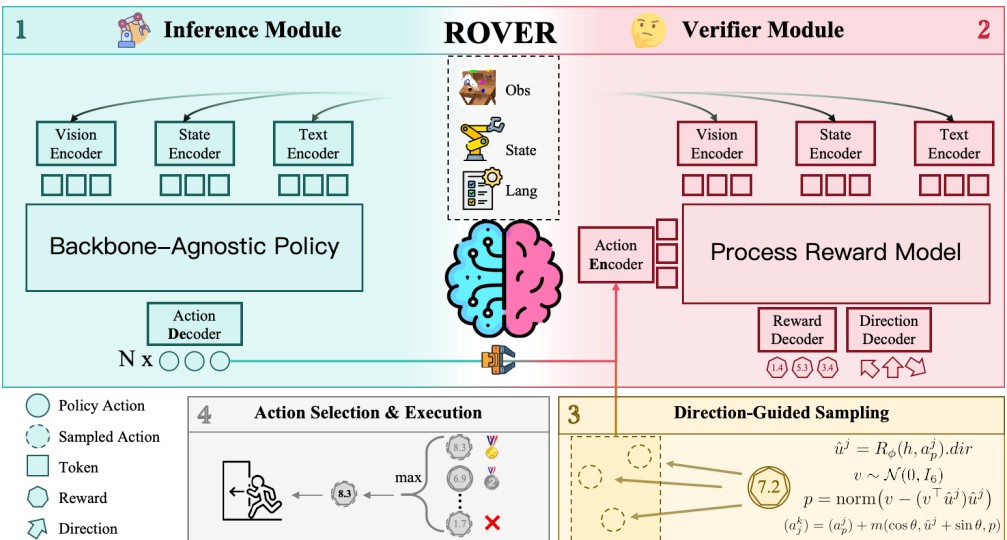

Figure 1: RoVer overview. A frozen VLA proposes $N$ actions; an external process reward model (PRM) scores candidates and predicts a refinement direction. Candidates are expanded along the guided direction and the top-scoring action executes. Perceptual features are cached once and reused across candidates to amortize compute.

advocates for strategically reallocating computational resources from extensive pre-training to the inference stage, leading to more efficient performance gains. Further strategies like self-consistency and Monte Carlo Tree Search also leverage PRMs to produce diverse and integrated outputs.

**Test-time Scaling in Vision-Language-Action Models.** *Internal test-time scaling:* Recent work increases a VLA's internal deliberation at inference: Embodied Chain-of-Thought(Zawalski et al., 2025; Ji et al., 2025; Sun et al., 2024; Zhou et al., 2025) approaches enforce multi-step reasoning before action generation to improve task decomposition and planning , often by expanding training datasets with reasoning annotations. CoT-VLA and UniVLA(Zhao et al., 2025; Bu et al., 2025b; Wang et al., 2025) incorporate future-frame prediction during inference; OneTwo-VLA(Lin et al., 2025) uses [BOR] and [BOA] tokens to adaptively decide when to reason and when to act. However, most of the above work needs additional annotation of the training dataset. *External test-time scaling:* In contrast to internal TTS, which reasons *within* the policy, external TTS *decouples* search and scoring from the policy by introducing a separate reward/value verifier that evaluates candidate actions at inference, guiding candidate generation without modifying backbone weights. Concurrent works explore this direction: Hume(Song et al., 2025) augments a dual-system VLA with a value head to drive repeated sampling and cascaded denoising (with fixed candidate counts), V-GPS(Nakamoto et al., 2024) uses a learned value function to select actions, and RoboMonkey(Kwok et al., 2025) studies reward modeling with large backbones and synthetic data at scale. Our work instantiates external TTS with a compact process reward model and systematically examines scaling with candidate budgets and compatibility across backbones; see Section 3.2.

## 3 METHOD

**Overview** We augment a frozen VLA with an external process reward model (PRM) that, given observations, language, and a candidate action, outputs a scalar score and a direction toward improvement. At inference, policy proposals are expanded by sampling along the predicted direction within an angular bound, and the top-scoring action under the PRM is executed. Observation, language, and state features are computed once per step and cached that is reused across candidates (see Fig. 1).

### 3.1 PRELIMINARY

**Policy Models** We consider sequential decision-making with observations $o_t$ (e.g., RGB-D images, proprioception) and a natural language goal $g$, and history $h_t = (o_{\leq t}, g)$ . A pre-trained VLA policy

$\pi_\theta$ maps history $h_t$ to a distribution over low-level actions $a_t \in A$:

$$\pi_\theta(a_t \mid h_t) \in \Delta(A), \tag{1}$$

We use an end-effector *delta* action with gripper command, $a_t = [\Delta p_t, \; \Delta q_t, \; g_t] \in \mathbb{R}^{d_a}$, with $d_a = 7$ in our settings ($\Delta p_t \in \mathbb{R}^3, \Delta q_t \in \mathbb{R}^3, g_t \in \mathbb{R}$). At inference, one may either execute $a_t = \arg\max_a \pi_\theta(a \mid h_t)$.

**Reward Models** To unlock additional capability at test time, we introduce a reward model that predicts (i) a scalar *process reward* and (ii) an *action-space direction* pointing from the current action toward a potentially better one:

$$R_\phi(h_t, a_t^i) \to r_t^i, d_t^i, \quad r_t^i \in \mathbb{R}, \; d_t^i \in \mathbb{R}^{d_{\mathrm{dir}}}. \tag{2}$$

We predict directions for the actionable subspace of the control (excluding discrete gripper when present). The dimension $d_{\mathrm{dir}}$ matches the action subspace; in practice we use the normalized direction $\hat{u}_t^i = \mathrm{normalize}(d_t^i)$.

**Notation** From this section onward, unless otherwise stated, we omit the time index $t$ for clarity. Subscripts denote action sources: $a^i$ for the $i$-th candidate, $a_e$ for the expert action from the training dataset, $a_p$ for the policy action, and $a_{\mathrm{anc}}$ for the anchor action sampled around $a_e$ during training.

**Test-time scaling** Given a base policy $\pi_\theta$ and a verifier $R_\phi$, we obtain the policy action $a_p$ from $\pi_\theta$ and expand it into a candidate set $\mathcal{A} = \{a^0 = a_p, a^1, a^2, ...\}$ by Gaussian noise sampling. The PRM verifier $R_\phi$ scores each candidate with $r^i = R_\phi(h, a^i)$, and we execute $a^\star = \arg\max_{a \in \mathcal{A}} r(h, a)$. We illustrate this framework in detail in Section 3.2.3.

### 3.2 RoVer

RoVer instantiates external test-time scaling with a compact process reward model (PRM) that scores candidate actions and predicts a refinement direction in the action subspace. For policies defined in local frames, actions are mapped to the world frame before expansion and scoring. The following subsections detail the architecture, training objective, and inference procedure.

#### 3.2.1 Model Architecture

The $R_\phi$ takes synchronized multi-modal inputs $o_t$ (e.g., third-person and eye-in-hand RGBs, robot states, and language tokens) together with a candidate action $a^i$, and outputs both a scalar process reward $r^i$ and an action-space direction $d^i$. The model architecture follows the GPT-2 style(Radford et al., 2019), and is initialized with the pre-trained weight of GR-1(Wu et al., 2023). Specifically, the image encoder is initialized from a MAE(He et al., 2021) pre-trained model, and the text encoder is initialized from the CLIP text encoder(Radford et al., 2021). The architecture in principle supports up to 10 timesteps of history as input. However, for better plug-and-play usage and faster inference, we restrict inputs to the current timestep observation. Additional heads for reward and direction prediction are added on top of the initialized backbone. Due to all candidate actions sharing the same observation, language, and state at a control step, we compute these perceptual features once and reuse them across candidates as a shared *perception cache*, while encoding actions per candidate to amortize computation. Compared to fine-tuning a 7B-parameter backbone in RoboMonkey(Kwok et al., 2025), RoVer takes 0.2B parameters in total and only 40M for training. See Fig. 2.

**Action Amplifier** To make small differences between candidate actions more discernible, we apply a lightweight *action amplifier* to the action embedding before fusing it with observation/language tokens. The amplifier is a compact MLP with GELU and LayerNorm mapping $\mathbb{R}^H \to \mathbb{R}^{2H} \to \mathbb{R}^H$, reconditioning the action channel so that fine-grained deltas in the action subspace remain salient under strong frozen perception/language backbones. This contrast enhancer improves the PRM's ability to discriminate and rank nearby actions while keeping inference overhead minimal. For intuition, the right panel of Fig. 2 visualizes the predicted 2D direction field on PushT when the underlying base policy outputs planar $(x, y)$ actions.

#### 3.2.2 Model Training

The training objective of the reward model $R_\phi$ is to equip it with the ability to distinguish which of two candidate actions is better. In RoVer, we regard one action as better if its root mean squared

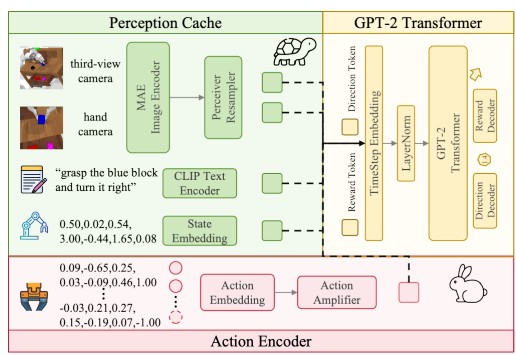 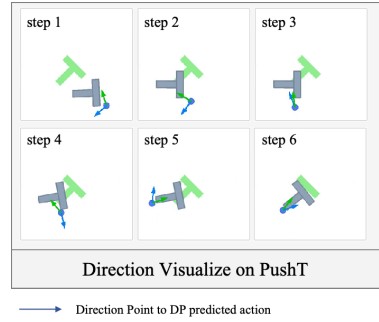

Figure 2: Left: verifier architecture. A shared perception cache and per-candidate action encoder (with an action amplifier) feed a GPT-2 backbone via reward/direction tokens, producing a scalar process reward and an action-space refinement direction. Right: direction visualization on the PushT benchmark, where PRM is trained atop a pre-trained Diffusion Policy and predicts 2D $(x, y)$ action directions; for CALVIN, the PRM predicts 6D pose directions. Note: on PushT, a DP pre-trained policy already achieves near-100% success; we primarily use PushT here for intuitive direction visualization rather than performance gains.

error(RMSE) distance to the expert action is smaller than that of the other. To prepare training data, we first analyze the distribution gap between policy action $A_\mathrm{p}$ and expert action samples $A_\mathrm{e}$ (Appendix A.2). Based on this analysis, we set a base noise scale $\sigma_\mathrm{base} = 0.1$, which is used to construct anchor actions $a_\mathrm{anc}$. Given expert demonstrations $h, a_\mathrm{e} = [a_\mathrm{e}^\mathrm{pose}, g_\mathrm{e}]$, we then construct local action tuples around $a_\mathrm{e}$ to obtain informative preference labels and direction supervision.

**Direction-guided and anchor-centered sampling** In early experiments, we observed that simply sampling noisy expert action $a_\mathrm{en}$ around $a_\mathrm{e}$ yielded poor performance. To better simulate the test-time sampling behavior, we introduce the notion of an *anchor action*. We form an anchor noise by perturbing the expert action in the 6D pose subspace (for CALVIN):

$$a_\mathrm{anc}^\mathrm{pose} = a_\mathrm{e}^\mathrm{pose} + \boldsymbol{n}, \qquad \boldsymbol{n} \sim \mathcal{N}(\mathbf{0}, \sigma_\mathrm{base}^2, I_6), \quad g_\mathrm{anc} = g_\mathrm{e}. \tag{3}$$

We then define the ground-truth direction vector from the anchor action to the expert action:

$$u_\mathrm{gt} = \frac{a_\mathrm{e}^\mathrm{pose} - a_\mathrm{anc}^\mathrm{pose}}{\left\| a_\mathrm{e}^\mathrm{pose} - a_\mathrm{anc}^\mathrm{pose} \right\|_2} \in \mathbb{R}^6. \tag{4}$$

Using $u_\mathrm{gt}$, we define the orthogonal hyperplane

$$\mathcal{H} = \{\, a \in \mathbb{R}^6 \mid (a - a_\mathrm{anc}^\mathrm{pose})^\top u_\mathrm{gt} = 0 \} \tag{5}$$

which partitions the space into two half-spaces:

$$\mathcal{H}^+ = \{\, a \in \mathbb{R}^6 \mid (a - a_\mathrm{anc}^\mathrm{pose})^\top u_\mathrm{gt} > 0 \,\}, \tag{6}$$

$$\mathcal{H}^- = \{\, a \in \mathbb{R}^6 \mid (a - a_\mathrm{anc}^\mathrm{pose})^\top u_\mathrm{gt} < 0 \,\}. \tag{7}$$

Let $d_0 = \sqrt{\frac{1}{6} \left\| a_\mathrm{e}^\mathrm{pose} - a_\mathrm{anc}^\mathrm{pose} \right\|_2^2}$ be the RMSE distance between $a_\mathrm{anc}$ and $a_\mathrm{e}$ in the 6D pose subspace. We define an adaptive noise scale as

$$\sigma_\mathrm{adapt} = \mathrm{clip}\big(k\, d_0,\ \sigma_\mathrm{min},\ \sigma_\mathrm{base}\big), \quad k > 0,\ \sigma_\mathrm{min} > 0, \tag{8}$$

to control the spread of candidate actions around $a_\mathrm{anc}$. In our implementation, we sample $\boldsymbol{\epsilon}_b, \boldsymbol{\epsilon}_w \sim \mathcal{N}(\mathbf{0}, \sigma_\mathrm{adapt}^2 I_6)$ and project them into the correct half-spaces:

$$\text{better: if } \boldsymbol{\epsilon}_b^\top u_\mathrm{gt} \le 0,\ \boldsymbol{\epsilon}_b \leftarrow -\boldsymbol{\epsilon}_b; \qquad \text{worse: if } \boldsymbol{\epsilon}_w^\top u_\mathrm{gt} \ge 0,\ \boldsymbol{\epsilon}_w \leftarrow -\boldsymbol{\epsilon}_w. \tag{9}$$

This yields

$$a_\mathrm{better}^\mathrm{pose} = a_\mathrm{anc}^\mathrm{pose} + \boldsymbol{\epsilon}_b, \quad a_\mathrm{worse}^\mathrm{pose} = a_\mathrm{anc}^\mathrm{pose} + \boldsymbol{\epsilon}_w \tag{10}$$

with the gripper state inherited from $a_\mathrm{anc}$. By construction, $a_\mathrm{better}$ is more likely to lie closer to the expert action than $a_\mathrm{anc}$, while $a_\mathrm{worse}$ is farther away.

**Supervision and Objective** From each anchor-centered tuple {anchor, better, worse}, we supervise two signals. For direction supervision, the ground-truth unit vector from a sampled action $a_\text{x}$ toward the expert is

$$u_\text{gt}(a_\text{e}, a_\text{x}) = \frac{a_\text{e}^\text{pose} - a_\text{x}^\text{pose}}{\left\| a_\text{e}^\text{pose} - a_\text{x}^\text{pose} \right\|_2}. \tag{11}$$

Let $\hat{u} = \text{normalize}\big(d_\phi(h, a_\text{x})\big)$ denote the predicted direction. We minimize the cosine misalignment

$$\mathcal{L}_\text{dir} = \mathbb{E}\big[\, 1 - \langle \hat{u},\ u_\text{gt} \rangle \,\big], \tag{12}$$

averaged over all sampled actions in the tuple. For reward supervision, let $r(a) = R_\phi(h, a)$ denote the PRM score. For every ordered pair $i \succ j$ within a tuple where $a_i$ is closer to the expert action than $a_j$, we apply the Bradley–Terry preference loss(Bradley & Terry, 1952):

$$\mathcal{L}_\text{rew}(i \succ j) = -\log \sigma\big(r(a_i) - r(a_j)\big), \tag{13}$$

and define $\mathcal{L}_\text{rew}$ as the average over all such pairs.

The final training objective combines both terms:

$$\mathcal{L}_\text{total} = \lambda_\text{dir}\mathcal{L}_\text{dir} + \lambda_\text{rew}\mathcal{L}_\text{rew}, \tag{14}$$

where $\lambda_\text{dir}$ and $\lambda_\text{rew}$ balance the two losses. We use standard gradient clipping during optimization. For validation, we track cosine alignment, angle error, and the monotonicity of PRM scores with respect to action–expert distance.

### 3.2.3 DIRECTION-GUIDED TEST-TIME SCALING

At inference time, RoVer augments the frozen VLA policy by expanding policy actions into a set of candidate actions and selecting the best one under the PRM. The key distinction from training is that candidates are generated from the policy action $a_\text{p}$ rather than from expert actions $a_\text{e}$, and only the "better" side of the action space is explored with the guide of predicted direction $\hat{u}$. We implement two sampling strategies and compare: **i)** Random sampling: perturb $a_\text{p}$ with Gaussian noise without guidance; **ii)** Direction-guided sampling: expand $a_\text{p}$ by sampling along the PRM-predicted direction.

This procedure converts additional test-time computation into improved action selection. Random sampling explores broadly but inefficiently, while direction-guided sampling exploits the PRM's predicted direction to concentrate candidates in promising regions, yielding better performance under the same budget.

## 4 EXPERIMENT

We evaluate RoVer both in simulation and on a real robot platform. For simulation, we adopt the CALVIN benchmark under the ABC→D setting: models are trained in three environments (A, B, C) and tested in an unseen environment (D). We select three representative state-of-the-art baselines from different stages of VLA research — GR-1(Wu et al., 2023), Dita(Hou et al., 2025), and MoDE(Reuss et al., 2025) — to demonstrate that RoVer consistently improves policies of varying capacity. For real-robot experiments, we use Diffusion Policy (DP)(Chi et al., 2024; 2023) as the base policy and show that RoVer also enhances performance in physical manipulation tasks.

### 4.1 CALVIN BENCHMARK EXPERIMENTS

CALVIN is a widely used simulation benchmark for long-horizon robot manipulation (Mees et al., 2022). It provides multi-stage environments with language-conditioned tasks, enabling evaluation of both skill composition and generalization to unseen configurations. The ABC→D split is particularly challenging since it requires models trained in three training environments (A, B, C) to transfer to a held-out environment (D).

**Experiment Setup:** For RoVer, we reuse the GR-1 backbone architecture as the verifier, replacing the original action token with a reward token and a direction token. The total parameter size is 0.2B, with only 40M parameters trainable – the rest are frozen from MAE and CLIP text encoders. The reward model is trained on the CALVIN ABC→D training split for 100 epochs, and

Table 1: CALVIN Benchmark results on the ABC→D split. Columns report probability of completing $k$ tasks in a row and the average chain length. All baseline numbers are taken from the official leaderboard (http://calvin.cs.uni-freiburg.de/), except for GR-1*. The GR-1 baseline is reported on the leaderboard with an average chain length of 3.06, but in our local reproduction we observed stronger performance (Avg. Len. 3.19); we therefore report the locally evaluated GR-1* as the base policy. † denotes the best performance achieved by applying RoVer test-time scaling on top of the corresponding base policy, and ‡ denotes applying the V-GPS-style test-time verifier(Nakamoto et al., 2024) to the corresponding base policy. For both RoVer and V-GPS, the Δ rows report absolute changes relative to the corresponding base policy.

| Method | 1 | 2 | 3 | 4 | 5 | Avg. Len. |
|---|---|---|---|---|---|---|
| GR-1* | 85.1 | 73.7 | 63.2 | 53.7 | 43.4 | 3.19 |
| GR-1‡ | 86.4 | 73.0 | 62.3 | 50.8 | 40.5 | 3.13 |
| Δ | +1.3 | -0.7 | -0.9 | -2.9 | -2.9 | -0.06 |
| 3D Diffuser Actor | **92.2** | **78.7** | 63.9 | 51.2 | 41.2 | 3.27 |
| GR-1† | 86.1 | 75.3 | **66.8** | **56.2** | **48.7** | **3.33** |
| Δ | +1.0 | +1.6 | +3.6 | +2.5 | +5.3 | +0.14 |
| Dita | 94.5 | 82.5 | 72.8 | 61.3 | 50.0 | 3.61 |
| Dita‡ | 93.2 | 83.2 | 72.8 | 64.4 | 53.2 | 3.67 |
| Δ | -1.3 | +0.7 | 0.0 | +3.1 | +3.2 | +0.06 |
| RoboUniView | 94.2 | 84.2 | 73.4 | 62.2 | 50.7 | 3.64 |
| GHIL-Glue | **95.2** | **88.5** | 73.2 | 62.5 | 49.8 | 3.69 |
| Dita† | 94.8 | 83.2 | **76.8** | **70.0** | **59.2** | **3.84** |
| Δ | +0.3 | +0.7 | +4.0 | +8.7 | +9.2 | +0.23 |
| MoDE | 96.2 | 88.9 | 81.1 | 71.8 | 63.5 | 4.01 |
| MoDE‡ | 96.4 | 89.9 | 82.1 | 72.8 | 65.5 | 4.07 |
| Δ | +0.2 | +1.0 | +1.0 | +1.0 | +2.0 | +0.06 |
| MoDE† | **97.1** | **90.9** | **82.5** | **74.9** | **66.6** | **4.12** |
| Δ | +0.9 | +2.0 | +1.4 | +3.1 | +3.1 | +0.11 |

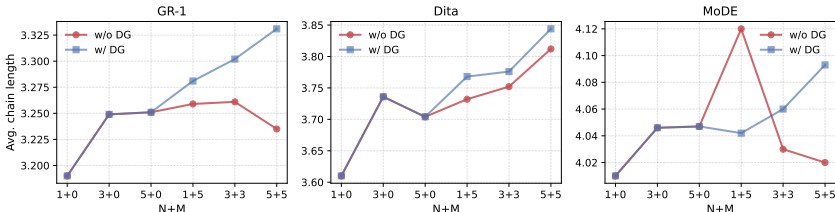

Figure 3: Direction-guided test-time scaling: average chain length versus $N+M$ ($N$ policy proposals, $M$ guided expansions). Panels correspond to GR-1, Dita, and MoDE. Red: unguided (random) expansion; blue: direction-guided (DG). MoDE shows broad but less stable gains due to the chunk–step mismatch discussed in Q2.

we use the final checkpoint for evaluation on all backbones. Importantly, we only sample 20% of the training set, showing that RoVer is also training-efficient. We study the following questions: **Q1 (Backbone-agnostic gains):** Does RoVer consistently improve different pre-trained baselines (GR-1, Dita, MoDE; and DP on real robot)? **Q2 (Scaling and sampling efficiency):** How does performance scale with the number of policy proposals $N$ and the guided expansion budget $M$ per proposal; under a fixed candidate budget $K=N+M$, is direction-guided sampling more effective than unguided Gaussian expansion? **Q3 (Inference efficiency):** Does a shared perception cache amortize computation and improve throughput/latency at test time?

**Baseline Methods:** We summarize the three baselines and highlight the test-time considerations when integrating them with RoVer. Unless otherwise noted, RoVer applies a unified candidate handling across backbones: (i) expansion noise is injected only into the 6D arm pose components; (ii) the gripper dimension is not noised and is selected via a simple vote across the $N$ base proposals; and (iii) perceptual features are pre-encoded once per step (pre_encode) and reused across candidates.

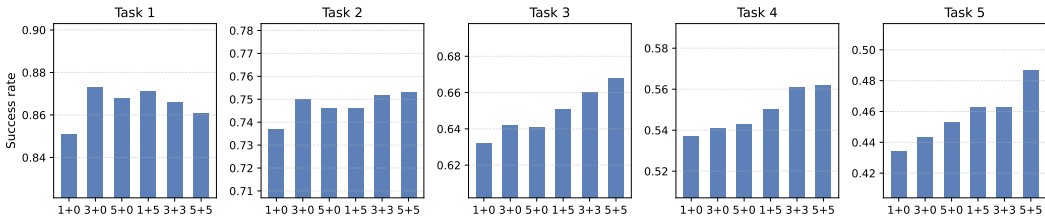

Figure 4: GR-1: SR@k ($k$=1 . . . 5) as a function of $N$ (policy proposals). Increasing $N$ boosts success across all $k$, revealing the value of proposal diversity; together with Fig. 3, this shows that adding direction-guided expansions $M$ on top of non-trivial $N$ yields further gains.

Table 2: Inference efficiency with/without a shared perception cache. Latency per control step decreases substantially when caching perception features once per step; speedup grows with the number of candidates.

| #Actions | w/o cache (s) | w cache (s) | Speedup ($\times$) | Per-action (ms) |
|---|---|---|---|---|
| 10 | 0.4140 | 0.0929 | 4.46 | 9.29 |
| 100 | 4.1400 | 0.6174 | 6.71 | 6.17 |
| 1000 | 41.4748 | 5.7384 | 7.23 | 5.74 |
| 10000 | 418.1795 | 58.2168 | 7.18 | 5.82 |

*GR-1* (Wu et al., 2023) is a GPT-style trajectory model that conditions on multi-view RGB and language to autoregressively predict end-effector deltas (6D arm pose delta plus a gripper scalar). At inference, we draw $N$ stochastic proposals per control step, optionally expand each action within a bounded-angle region around the PRM-predicted direction, and rank all candidates by the PRM. Because GR-1 natively operates in world-frame deltas, no additional frame conversion is required.

*Dita* (Hou et al., 2025) is a Diffusion Transformer policy that predicts relative motions in the local end-effector frame (position and Euler-angle deltas), which are converted to world-frame deltas using the current end-effector pose. Under RoVer, we first sample $N$ policy actions in the local frame, convert each to a world-frame action, and then perform noise expansion in the world frame to align with the PRM's world-frame direction guidance.

*MoDE* (Reuss et al., 2025) is a diffusion-transformer with Mixture-of-Expert denoisers that outputs a short action chunk (a sequence of future actions). During test-time scaling with RoVer, we draw multiple candidate chunks and interface with the PRM by scoring the first action of each chunk (optionally after direction-guided expansion of its 6D arm components) to choose the executed action. Alternative chunk-execution strategies are orthogonal and omitted here.

**Q1: Backbone-agnostic gains** Without any retraining of the base policies, plugging the same RoVer verifier into different backbones yields consistent improvements (Table 1). For average chain length, GR-1 improves from 3.19 to 3.33, Dita from 3.61 to 3.84 and MoDE from 4.01 to 4.12. Looking at long-horizon success (SR@5), GR-1 rises from 41.5% to 48.7% (+17.4%), while Dita goes from 50.0% to 59.2% (+18.4%).

**Q2: Direction-guided test-time scaling** We study scaling under a unified view of compute: given a total candidate budget $K$=$N$+$M$ (policy proposals $N$ and direction-guided expansions $M$), performance generally increases with $K$, and direction guidance (DG) further improves sample efficiency by focusing exploration. Empirically (Figs. 3 and 4), increasing $N$ first yields large gains at small budgets by diversifying policy modes; adding a modest $M$ then refines promising proposals and yields additional improvements. Under equal $K$, DG reliably outperforms unguided Gaussian expansion on GR-1 and Dita, consistent with the intended role of the direction head.

For MoDE, however, gains are not consistently higher across all settings (right panel of Fig. 3). We attribute this to a chunk–step mismatch: MoDE outputs short action chunks, whereas our PRM is trained and applied per time step. To maintain a unified evaluation pipeline, we expand and score only the first action of each chunk and then execute the selected chunk; subsequent steps inside the chunk receive no further guidance. This limits our ability to intervene within chunks and can produce non-monotonic trends as $N$+$M$ increases. Nevertheless, the method remains effective—most $N$+$M$ settings still show positive improvements.

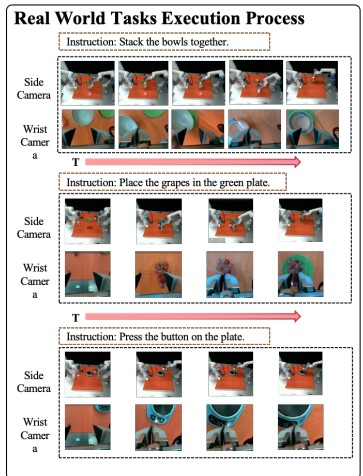 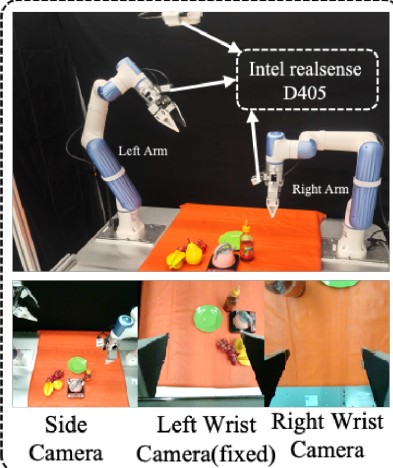

Figure 5: Left: visualization of our real-robot experiments. Right: the dual-arm Dobot testing platform used in our evaluation.

Table 3: Real-robot results: success rate (%). Each entry is computed over 10 trials per condition.

| Method | Pick and place | | | Push button | | Stack bowls | | Avg. |
|--------|------|---------------|-----------------|------|-----------------|------|-----------------|------|
| | Seen | Unseen object | Unseen position | Seen | Unseen position | Seen | Unseen position | |
| DP | 100% | 90% | 50% | 100% | 50% | 80% | 40% | 72.9% |
| DP† | 100% | 100% | 70% | 100% | 80% | 100% | 70% | 88.6% |

**Q3: Effect of a shared perception cache** We evaluate the impact of caching perceptual features across candidates by measuring wall-clock latency per control step under increasing candidate budgets. As shown in Table 2, caching reduces per-step latency substantially and the speedup grows with the number of candidates: for 10 candidates, latency drops from 0.414s to 0.0929s ($4.46\times$); for 100 and 1000 candidates, the speedups reach $6.71\times$ and $7.23\times$, respectively. With caching, the per-action cost stabilizes around 5.7–6.2 ms (e.g., 6.17 ms at 100 candidates and 5.74 ms at 1000 candidates), enabling near-linear scaling in the number of candidates for a single perception encode. These results validate that amortizing perception via a shared perception cache is critical to test-time scaling efficiency. All latency measurements are conducted on a single NVIDIA V100 GPU.

## 4.2 REAL ROBOT EXPERIMENTS

We evaluate RoVer on a dual-arm Dobot platform (Fig. 5). Since our tasks are single-arm, the left arm remains stationary while the right arm executes. Perception uses both an eye-in-hand wrist camera on the right arm and an overhead third-person camera. We consider three manipulation tasks—pick-and-place, push button, and stack bowls—each evaluated under **Seen**, **Unseen object**, and **Unseen position** conditions. Concretely: pick-and-place requires grasping a small household item and placing it at a designated location; push button requires moving to and actuating a panel button; stack bowls requires placing one bowl atop another with correct alignment. As shown in Table 3, RoVer (DP†) matches DP on seen cases while providing clear gains in generalization.

## 5 CONCLUSION

RoVer is an external test-time scaling framework that upgrades frozen VLA policies by introducing a compact process reward model to score and direction-guide candidate actions. Without retraining the backbones, RoVer consistently improves performance across diverse policies (Q1), scales effectively with candidate budgets—especially under direction guidance (Q2)—and achieves substantial inference speedups via a shared perception cache (Q3). While limitations remain (e.g., step–chunk mismatches for chunked policies and the use of expert proximity as a supervision proxy), RoVer demonstrates that reallocating compute from training to inference can reliably unlock additional capability in embodied policies without extra pre-training or data.

**Ethics Statement**

Our work involves only robotic simulation and real-robot experiments, and does not include human or animal subjects. All datasets are collected from robotic manipulation trials and contain no personally identifiable information. The methods proposed are intended for academic research on embodied AI and robotics. We are not aware of direct misuse risks, but as with any general-purpose learning method, caution is advised if deployed in safety-critical or adversarial settings.

**Reproducibility Statement**

We provide detailed descriptions of model architectures, training procedures, and evaluation protocols in the main paper and appendix. Hyperparameters, network sizes, and task setups are specified in the text and tables. Additional training details, data processing steps, and evaluation scripts will be open-sourced after acceptance.

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

# A    APPENDIX

## A.1    THE USE OF LARGE LANGUAGE MODELS

Large language models (LLMs) were not involved in the design, implementation, or evaluation of the proposed methods. Their sole use in this work was for minor linguistic assistance, such as polishing the writing style and improving readability of the manuscript. All technical ideas, experiments, and analyses were conceived and conducted entirely by the authors.

## A.2    ACTION DISTRIBUTION ANALYSIS

To calibrate the perturbation scale used by the preference reward model (PRM) and our test-time sampling, we quantified the distribution gap between policy actions $A_{\mathrm{p}}$ and expert actions $A_{\mathrm{e}}$ on the CALVIN validation set. Following the analysis script in GR-1 (visualizing 3D deviations and their projections), we collect per-step deviations $\Delta a = A_{\mathrm{p}} - A_{\mathrm{e}}$ and summarize both the coordinate-wise statistics for translation (XYZ) and the Euclidean distance $\|\Delta a_{xyz}\|_2$.

Key findings from the aggregated statistics are:

- Near-zero bias: mean translation deviation is small in all axes (X/Y/Z: 0.0009/0.0048/-0.0013 m). Medians are similarly close to zero.
- Anisotropic spread: standard deviations are (X/Y/Z: 0.184/0.116/0.165 m), indicating a slightly tighter spread along Y and broader tails along X and Z.
- Heavy-tailed distances: the Euclidean distance has mean 0.215 m and median 0.170 m with a long tail (min 0.003 m, max 1.708 m). This matches the visualizations where most samples cluster near the origin with a small fraction of large outliers.

Implications for training and sampling. We set a conservative base noise scale of $\sigma_{\mathrm{base}} = 0.1$ for constructing anchor-centered pairs: it is smaller than the median policy–expert gap (0.17 m), which (i) keeps most synthesized "better/worse" pairs on-manifold around policy proposals and (ii) avoids over-penalizing the verifier with rare, far-out outliers. At test time, the same scale serves as the initial radius for candidate expansion around policy actions; we combine it with direction guidance from the PRM to bias samples toward the expert manifold while preserving diversity.

Together with the qualitative 3D scatter and projection plots, this analysis supports our choice of a small but non-trivial perturbation radius and motivates the direction-aware, anchor-centered sampling used throughout the paper.

## A.3    ADDITIONAL REAL-ROBOT EXPERIMENTS WITH $\pi 0$

We evaluate RoVer on a dual-arm platform using $\pi 0$ as the base policy. We consider three manipulation tasks: **Pour water**, **Tidy up the desk**, and **Cook the vegetable**. Each task we collected 200 episodes for $\pi 0$ finetuning and PRM training. For *Pour water*, the right arm first grasps a bowl and brings it close to a kettle, while the left arm grasps and tilts the kettle to pour water into the bowl. For *Tidy up the desk*, the robot is asked to place multiple household objects that are randomly scattered on the table into a plastic basin. For *Cook the vegetable*, the robot must grasp a small toy vegetable, place it into a toy pot, and then grasp and place a small lid on top; due to the small graspable areas on both the vegetable and the lid, this task is substantially more challenging and yields lower success rates.

Figure 8 shows representative rollouts for *Pour water*, while Figures 9 and 10 visualize *Tidy up the desk* and *Cook the vegetable*, respectively. For the first two tasks, we display both the normal setting with the orange tablecloth used during teleoperation and the background-generalization setting with tablecloth removed; for *Cook the vegetable*, we show a normal rollout highlighting the fine-grained grasping challenges.

Furthermore, on *Tidy up the desk* we evaluate **novel-object generalization**: in the teleoperated dataset used to fine-tune $\pi 0$ and train the PRM, only starfruit and grapes are present; at test time, we add a bowl and a kettle as unseen objects and reuse the same language instructions and trained model.

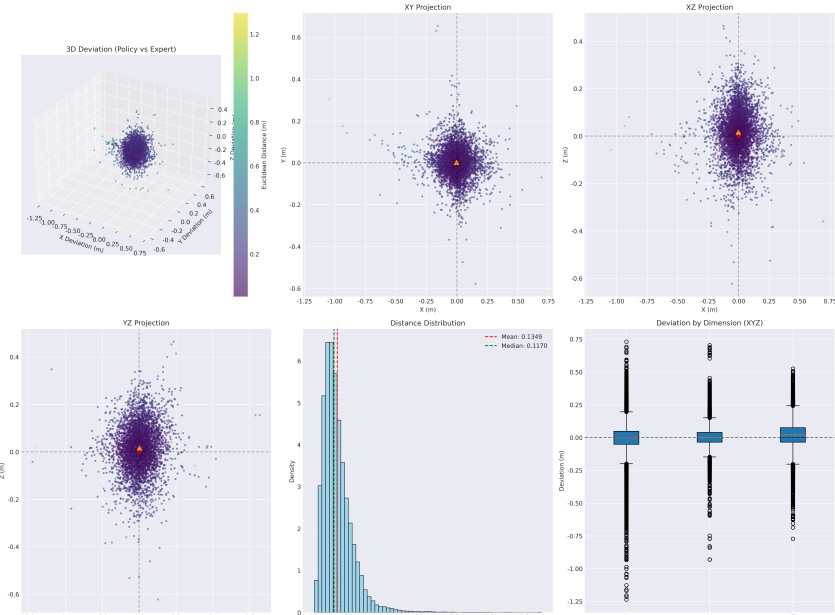

Figure 6: The visualization of distribution gap between policy action from GR-1 and expert action.

Two observations emerge. First, in-distribution settings (*Normal*), RoVer consistently improves success rates over $\pi 0$ across all three tasks, despite using only a modest number of candidates per step. Second, RoVer exhibits a conservative "safety" behavior on out-of-distribution settings. In the *Background* and *New obj* conditions, the scenes or objects are unseen for the PRM, whereas $\pi 0$ has been pre-trained on diverse visual data. In these cases, we frequently observe that RoVer assigns the highest reward to the original base action and rarely chooses perturbed candidates, effectively falling back to $\pi 0$ and preserving its generalization performance. This mirrors the behavior seen in the noise-scale ablation (Table 4): when the verifier is uncertain or perturbations are clearly suboptimal, RoVer prefers not to override the base policy, providing a practical lower bound on performance.

## A.4   ROVER TRAINING AND INFERENCE ALGORITHMS

For completeness, we summarize RoVer's full inference and training pipelines in Algorithm 1 and Algorithm 2. The inference procedure describes how we draw multiple stochastic proposals from a frozen base policy, aggregate gripper decisions, expand each proposal along the PRM-predicted direction in the continuous action subspace, and then select the final action via PRM-scored candidate ranking. The training procedure details our anchor-centered adaptive sampling scheme: starting from expert actions, we construct anchor, better, and worse actions with an adaptive noise scale around the anchor and supervise both the reward and direction heads using Bradley–Terry preference losses and cosine alignment.

## A.5   NOISE-SCALE ABLATION

To study the robustness of RoVer to the test-time noise scale used for direction-guided expansion, we vary the standard deviation $\sigma$ of the Gaussian perturbation around each GR-1 policy proposal on CALVIN ABC$\rightarrow$D while keeping the candidate budget fixed to $N{=}5$ and $M{=}5$. As shown in Table 4, RoVer achieves strong performance across a broad range of noise scales, with the best average chain length at a moderate $\sigma{=}0.05$. Larger noise scales do not catastrophically degrade performance: the PRM tends to assign low scores to clearly unreasonable candidates and fall back to the base proposals.

---

**Algorithm 1:** RoVer test-time scaling (inference)

---

**Input:** Observation $h$, base policy $\pi_\theta$, verifier $R_\phi$, number of policy actions $N$, expansions per action $K$, noise scale $\sigma$, angular bound $\alpha$

**Output:** Final executed action $a^\star$

**1. Policy step:** Query the base policy for $N$ stochastic actions
$\{a_p^1, a_p^2, \ldots, a_p^N\} \leftarrow \pi_\theta(h)$

**2. Gripper aggregation:** Extract the gripper component $g^j$ of each $a_p^j$ and compute a voted gripper $g^{\text{vote}}$ (e.g., majority over discrete open/close).

**3. Direction prediction on policy actions:** For each $j = 1, \ldots, N$
$(r_p^j, \hat{u}^j) \leftarrow R_\phi(h, a_p^j)$
Add $(a_p^j, r_p^j)$ to the candidate set $\mathcal{A}$.

**4. Direction-guided expansion:** For each $j = 1, \ldots, N$ and $k = 1, \ldots, K$
Sample a perturbation $\epsilon_{j,k}$ around direction $\hat{u}^j$ with magnitude controlled by $\sigma$ and cone angle $\alpha$.
Form expanded action $\tilde{a}_{j,k}$ by adding $\epsilon_{j,k}$ to the continuous arm-pose components of $a_p^j$ and setting its gripper to $g^{\text{vote}}$.
$(r_{j,k}, \_) \leftarrow R_\phi(h, \tilde{a}_{j,k})$
Add $(\tilde{a}_{j,k}, r_{j,k})$ to $\mathcal{A}$.

**5. Action selection:**
$a^\star \leftarrow \arg\max_{a \in \mathcal{A}} r(a)$

**return** $a^\star$

---

**Algorithm 2:** Training RoVer's PRM with anchor-centered adaptive sampling

---

**Input:** Dataset of expert transitions $\mathcal{D}$, base noise scale $\sigma_{\text{base}}$, scale factor $k$, minimum noise $\sigma_{\text{min}}$, loss weights $\lambda_{\text{dir}}, \lambda_{\text{rew}}$

**Output:** Trained verifier $R_\phi$

**for** *each training step* **do**

  Sample $(h, a_e) \sim \mathcal{D}$ ;                  // $a_e = [a_e^{\text{pose}}, g_e]$

  Sample $n \sim \mathcal{N}(0, \sigma_{\text{base}}^2 I_6)$ and set $a_{\text{anc}}^{\text{pose}} = a_e^{\text{pose}} + n$, $g_{\text{anc}} = g_e$ ;

  Compute $u_{\text{gt}}(a_e, a_{\text{anc}})$ and $d_0 = \sqrt{\frac{1}{6} \|a_e^{\text{pose}} - a_{\text{anc}}^{\text{pose}}\|_2^2}$ ;

  Set $\sigma_{\text{adapt}} = \text{clip}(k\, d_0, \ \sigma_{\text{min}}, \ \sigma_{\text{base}})$ ;

  Sample $\epsilon_b, \epsilon_w \sim \mathcal{N}(\mathbf{0}, \sigma_{\text{adapt}}^2 I_6)$ and enforce half-spaces $\epsilon_b^\top u_{\text{gt}} \geq 0$, $\epsilon_w^\top u_{\text{gt}} \leq 0$ ;

  Set $a_{\text{better}}^{\text{pose}} = a_{\text{anc}}^{\text{pose}} + \epsilon_b$, $a_{\text{worse}}^{\text{pose}} = a_{\text{anc}}^{\text{pose}} + \epsilon_w$ ;

  For each $a_x \in \{a_{\text{anc}}, a_{\text{better}}, a_{\text{worse}}\}$, compute $(r(a_x), \hat{u}(a_x)) = R_\phi(h, a_x)$ ;

  Update $\phi$ by minimizing $\mathcal{L}_{\text{total}} = \mathcal{L}_{\text{dir}} + \mathcal{L}_{\text{rew}}$ ;

---

## A.6 EFFECT OF DIRECTION-GUIDED SUPERVISION AT TRAINING TIME

Although RoVer's direction head is primarily used for test-time expansion, we find that direction-guided (DG) supervision also provides benefits even when inference-time DG is disabled. To isolate this effect, we train GR-1 with and without the direction loss (setting $\lambda_{\text{dir}}=0$ for the "w/o DG" variant) while keeping the reward loss identical, and evaluate both variants under the same candidate configurations $(N+M)$ using *only* unguided sampling at test time. As summarized in Table 5, DG supervision consistently improves long-horizon performance, especially at larger candidate budgets.

In addition to end-of-training success rates, we also monitor several validation metrics during PRM training that capture how well the learned verifier orders actions by quality. Concretely, on a held-out set of anchor-centered tuples we compute: *top-1 accuracy*, i.e., the probability that the expert action itself receives the highest score among the expert and its noisy perturbations; *mean reciprocal rank* (MRR), i.e., the average reciprocal rank of the expert action under the PRM-induced ordering; and *ranking consistency*, i.e., the fraction of tuples for which the noisy actions are ranked by PRM scores in the same order as their RMSE distances to the expert. These quantities are only proxies and do

Table 4: Effect of test-time noise scale $\sigma$ for GR-1 with RoVer on CALVIN ABC→D. Columns report probability of completing $k$ tasks in a row and the average chain length.

| Noise Ablation | 1 | 2 | 3 | 4 | 5 | Avg. Len. |
|---|---|---|---|---|---|---|
| 0.01 | **88.3** | **75.7** | 65.3 | 55.3 | 45.6 | 3.30 |
| 0.05 | 86.1 | 75.3 | **66.8** | **56.2** | **48.7** | **3.33** |
| 0.10 | 88.0 | 74.9 | 64.1 | 52.8 | 42.7 | 3.23 |
| 0.15 | 86.7 | 74.1 | 63.1 | 51.6 | 42.5 | 3.18 |
| 0.25 | 87.9 | 73.9 | 63.2 | 52.5 | 43.3 | 3.21 |
| 0.30 | **88.3** | 75.4 | 63.8 | 52.6 | 43.4 | 3.24 |

Table 5: Effect of direction-guided supervision (DG) during training on GR-1 performance on CALVIN ABC→D. All rows use unguided sampling at test time (no DG). Each block compares models trained without and with the direction loss under the same candidate configuration $N+M$.

| GR-1 $N+M$ | 1 | 2 | 3 | 4 | 5 | Avg. Len. |
|---|---|---|---|---|---|---|
| 3+0 | **87.5** | 74.0 | 64.2 | **54.5** | **44.7** | 3.249 |
| | 87.3 | **75.0** | 64.2 | 54.1 | 44.3 | 3.249 |
| 5+0 | **87.3** | 74.4 | 63.3 | 53.2 | 43.1 | 3.213 |
| | 86.8 | **74.6** | **64.1** | **54.3** | **45.3** | **3.251** |
| 1+5 | **87.5** | **74.4** | 63.7 | 53.1 | 42.9 | 3.216 |
| | 86.6 | 74.3 | **64.2** | **54.7** | **46.1** | **3.259** |
| 3+3 | 87.3 | 74.3 | 63.1 | 52.8 | 42.1 | 3.196 |
| | **87.4** | **74.5** | **64.5** | **54.4** | **45.3** | **3.261** |
| 5+5 | **87.7** | 73.6 | 62.8 | 52.7 | 43.4 | 3.202 |
| | 86.5 | **73.8** | **64.5** | **53.8** | **44.9** | **3.235** |

not directly determine the final test-time scaling behavior, but they provide a convenient diagnostic of training efficiency and the quality of the learned preference ordering.

Figure 7 plots these metrics over training steps for PRMs trained with and without DG supervision. Across all three measures, the DG-trained model achieves consistently higher validation performance, supporting the conclusion that direction supervision strengthens the underlying verifier even when test-time sampling does not explicitly use the predicted direction.

## A.7 IMPLEMENTATION DETAILS

**Stabilizing training with an Action Amplifier** During early training of the PRM, we observed a degeneration phenomenon where the reward head collapsed toward nearly constant outputs across inputs and the loss plateaued. Our hypothesis is that a strong frozen backbone (initialized from GR-1) dominated the feature fusion, causing small differences in action embeddings to be attenuated and thus providing weak gradients to the reward head. We therefore introduced an *Action Amplifier* — a lightweight MLP $H \to 2H \to H$ with GELU and LayerNorm — placed after the action embedding and before token fusion. This reconditions the action channel, preserves fine-grained deltas among nearby candidates, and restores useful gradients. After adding the amplifier, training became stable and the loss decreased reliably.

**Anchor-based dynamic sampling for robust preference learning** Using a single fixed noise scale to generate noisy action pairs around expert actions yielded a narrow training distribution and suboptimal test-time scaling. To diversify supervision while matching test-time behavior, we adopt an *anchor-centered* dynamic scheme (see Section 3.2): (i) sample an anchor action by adding base Gaussian noise in the 6D pose subspace to the expert action; (ii) compute the ground-truth direction $u_{\text{gt}}$ from the anchor toward the expert and define the orthogonal hyperplane; (iii) sample additional actions on the two half-spaces ("better" and "worse") using an adaptive noise scale clipped by the anchor–expert distance; and (iv) train with a Bradley–Terry preference loss over ordered pairs (closer-to-expert preferred) together with a direction loss that aligns predicted directions to $u_{\text{gt}}$. In

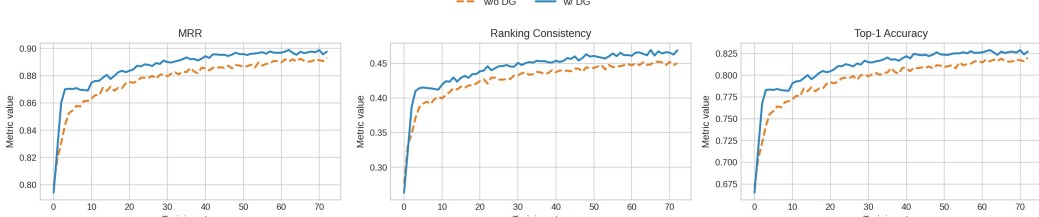

Figure 7: Validation metrics over training steps for GR-1 PRMs trained with (blue) and without (orange) direction-guided supervision. Left to right: mean reciprocal rank (MRR), ranking consistency, and top-1 accuracy on held-out anchor-centered tuples.

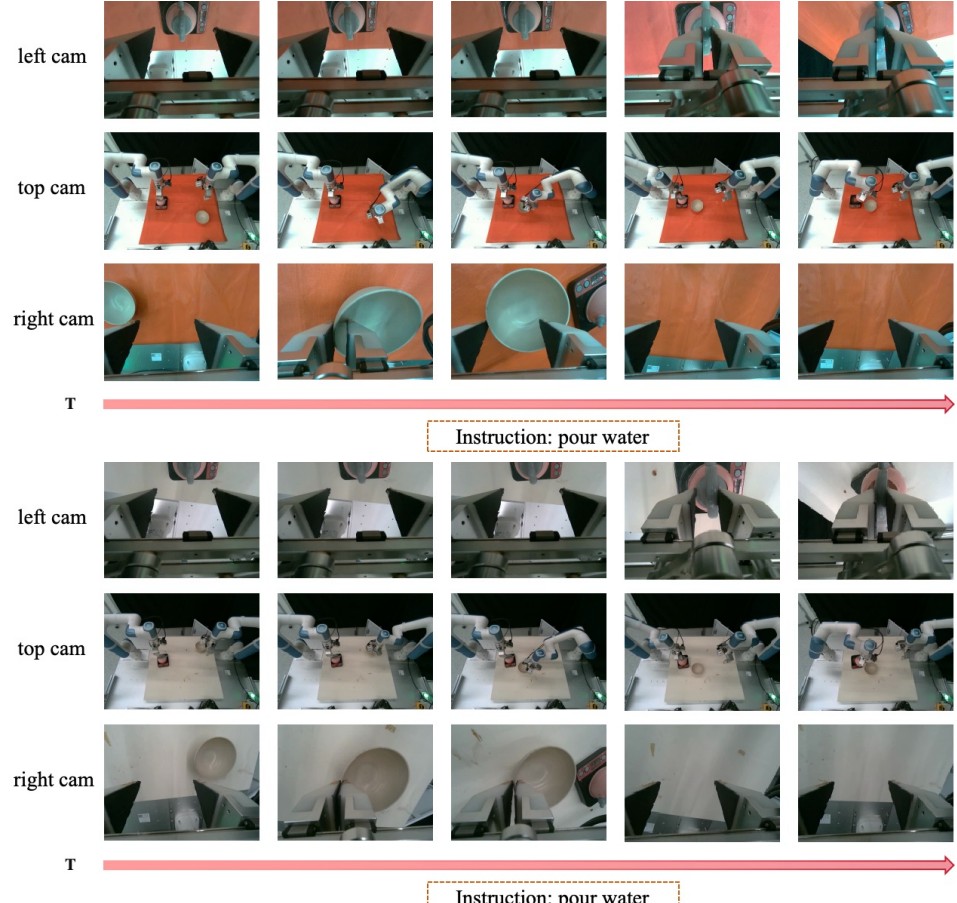

Figure 8: Representative rollouts for the *Pour water* task on the dual-arm platform with $\pi 0$ and RoVer. Top: normal setting with the orange tablecloth used during data collection. Bottom: background-generalization setting, where the tablecloth is removed and the robot acts on the same tabletop with white background.

practice, this produces a PRM that generalizes better to direction-guided TTS than models trained with fixed-scale Gaussian pairs.

**Model Architecture** The verifier $R_\phi$ is a lightweight GPT-2–style transformer with frozen CLIP text and MAE vision encoders. A Perceiver Resampler compresses image tokens; robot state and the 7D action are linearly embedded, and an *action amplifier* MLP ($H \to 2H \to H$) highlights small action differences. Two learned query tokens produce a scalar process reward and a normalized direction in the action subspace (typically 6D on CALVIN, 2D on PushT). For efficiency, we split computation into a shared 'pre_encode' step (perception/language/state) and a fast 'action_encode' step (per candidate), enabling near-linear scaling in the number of candidates. Key hyperparameters are summarized in Table 7.

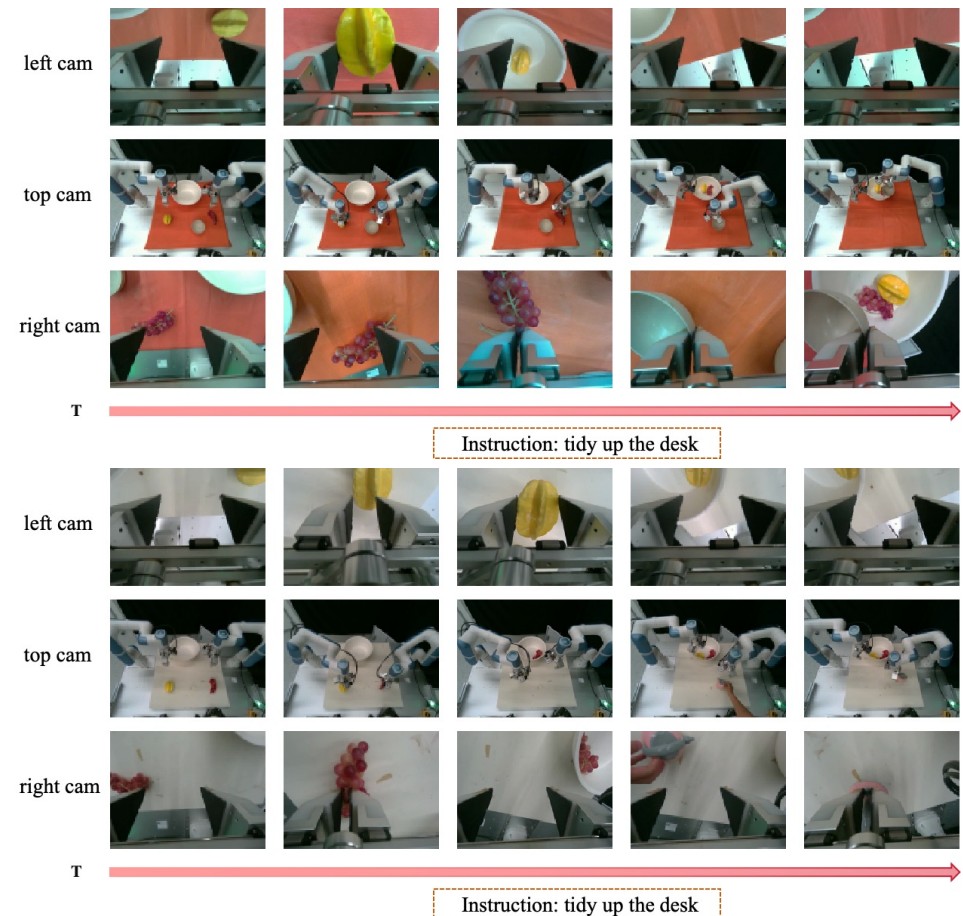

Figure 9: Representative rollouts for the *Tidy up the desk* task. Top: normal setting with scattered objects on the orange tablecloth. Bottom: background-generalization setting with the tablecloth removed.

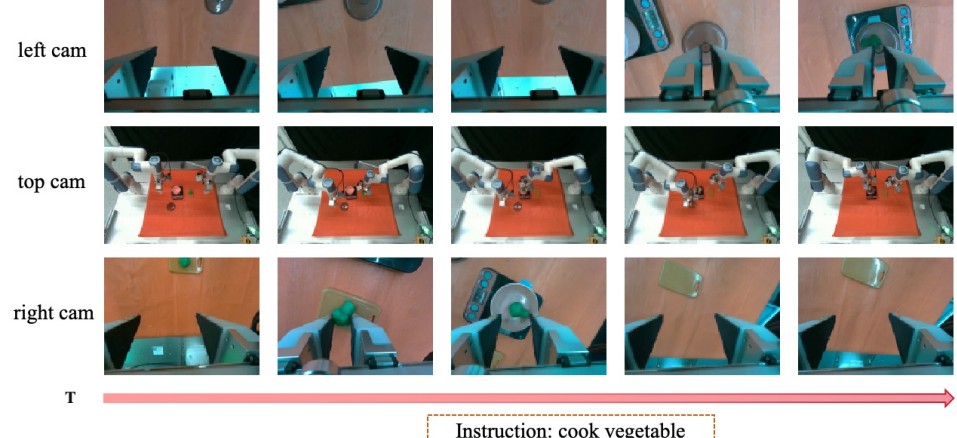

Figure 10: Representative rollout for the *Cook the vegetable* task in the normal setting. The robot must precisely grasp a small toy vegetable and lid, making this task substantially more difficult than the others.

## A.8  CALVIN BENCHMARK

We evaluate on the CALVIN benchmark (Mees et al., 2022) under the standard ABC→D generalization setting: we train reward model on environments A, B, and C, and evaluate on unseen

Table 6: Real-robot results with $\pi 0$ as the base policy. Each cell reports number of successful trials over total attempts. $\pi 0^\dagger$ denotes pairing $\pi 0$ with RoVer.

| Condition / Metric | Pour water | | Tidy up the desk | | Cook the vegetable | |
|---|---|---|---|---|---|---|
| | $\pi 0$ | $\pi 0^\dagger$ | $\pi 0$ | $\pi 0^\dagger$ | $\pi 0$ | $\pi 0^\dagger$ |
| Normal | 50% | 80% | 73.3% | 86.7% | 30% | 50% |
| Background | 50% | 50% | 53.3% | 60% | – | – |
| New obj | – | – | 46.7% | 46.7% | – | – |

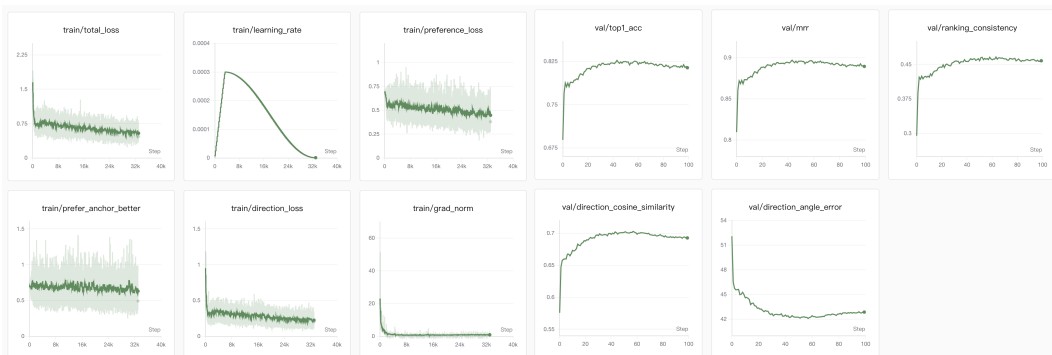

Figure 11: Training (left) and validation (right) logs for the PRM. With the Action Amplifier, losses decrease stably and the reward head avoids degenerate (constant) outputs.

environment D. CALVIN features long-horizon, language-conditioned tabletop manipulation with multi-object scenes and compositional goals. Each episode is paired with a natural-language instruction; success is counted when the instructed behavior is completed within a fixed horizon (see Fig. 12).

The task set covers rotation and pushing of colored blocks, opening/closing the drawer and slider, lifting/placing/stacking/unstacking blocks, toggling a lightbulb/LED, and pushing a block into the drawer. Following the official evaluation protocol of each backbone, we execute 1,000 multi-step sequences of 5 language instructions per sequence for GR-1 and MoDE, and 250 sequences for Dita. For each subtask, we allow up to 360 control steps and count success when the task-specific oracle detects completion.

Table 7: Verifier architecture hyperparameters (key = value).

| Key | Value |
| --- | --- |
| Hidden size | $H = 384$ |
| Layers | 12 |
| Attention heads | 12 |
| FFN | $4H$ |
| Dropout | 0.1 |
| Positions | 1024 |
| Seq. length | $L = 10$ |
| Text encoder dim | $512 \rightarrow H$ |
| Vision patch size | 16 |
| Resampler latents | 9 |
| Resampler depth | 3 |
| Resampler dim_head | 128 |
| Resampler heads | 4 |
| State dim | 6 (pose) + 2 (one-hot gripper) |
| Action dim | 7 |
| Amplifier | $H \rightarrow 2H \rightarrow H$ |
| Reward head | $H \rightarrow H/2 \rightarrow H/4 \rightarrow 1$ |
| Direction head | $H \rightarrow H/2 \rightarrow d_{\mathrm{dir}}$ |

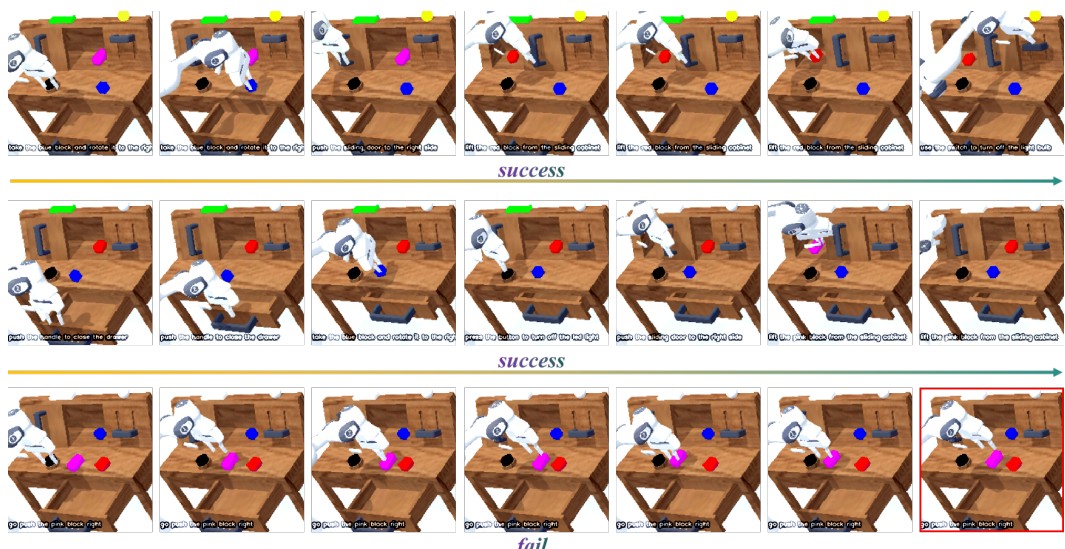

Figure 12: Overview of the CALVIN benchmark and the ABC→D split.

