# OpenReview forum: "RoVer: Robot Reward Model as Test-Time Verifier for Vision-Language-Action Model"
_ICLR.cc/2026/Conference — Submitted to ICLR 2026_

### Official Review · Reviewer_xijL · 2025-10-29

**Soundness:** 2
**Presentation:** 3
**Contribution:** 2
**Rating:** 4
**Confidence:** 3

**Summary:**

This paper introduce RoVer, a framework for test-time scaling in VLA models for robotics. It uses a PRM as verifier to evaluate and refine candidate actions from a frozen base policy without changing the model weights or needing more training data. The PRM gives scalar rewards and directions in action space to guide sampling better candidates. They cache perception features to make it efficient. Experiments on CALVIN sim and real robot show improvements over baselines like GR-1, Dita, MoDE, and Diffusion Policy.

**Strengths:**

1. Novel idea to shift scaling from training-time to test-time, which is important because robot data is expensive to collect. This plug-and-play approach can work with existing VLAs without retraining, that is practical.

2. The PRM design is clever, predicting both reward and direction allow for guided exploration instead of just random sampling. The action amplifier and anchor-centered training seem to help distinguish fine action differences.

3. Efficiency with shared perception cache is good, allow evaluating more candidates under same compute budget. Ablations in the method section show direction-guided sampling outperform random.

4. Experiments cover sim (CALVIN ABC->D) and real robot, with multiple baselines. Consistent gains, like improving success rates on long-horizon tasks.

**Weaknesses:**

1. The training of PRM still require the same dataset as the base policy, so it's not completely "without additional data" as claimed – it's reusing data but for a separate model. Could be clearer on this.

2. Limited to delta actions in end-effector space; not sure how it generalize to other action representations like joint angles or velocity control.

3. Real robot experiments only with Diffusion Policy, would like see on more advanced VLAs. Also, no latency numbers for inference – how much slower is it with N=64 candidates?

4. Direction prediction visualized on PushT, but PushT is easy (near 100% for DP), so gains are small. More analysis on when direction helps vs fails would be nice.

**Questions:**

1. In training, why choose RMSE to expert action as "better"? Is this process reward correlate well with actual task success? Maybe compare to outcome rewards.

2. For inference, how sensitive to noise scales σ_base and σ_adapt? Any hyperparam tuning details?

3. Compatibility: does RoVer work better with some base policies than others? E.g., diffusion vs transformer policies.

4. Appendix mentioned for distribution gap analysis, but not provided; can authors include more details?

---

> ### Author Response · Authors · 2025-11-24
>
> > $\color{red}{W1:}$ PRM data requirements and reuse
>
> Thank you for the careful review and for pointing this out. You are correct that training the PRM in RoVer still relies on the same training data as the policy. However, current VLA models typically require a few-shot fine-tuning stage on real-robot data before deployment. Thus, in practice, this data that we “reuse” for the PRM can be collected naturally during real-world deployment. What we would like to emphasize is that, unlike RoboBrain[1], we do not require additional manual annotations on top of these trajectories. Of course, we also look forward to future work that explores large-scale pre-training paradigms for PRMs with stronger generalization.

---

> ### Author Response · Authors · 2025-11-24
>
> > $\color{red}{W2 \\& W3:}$ Action space limits and real-robot baselines
>
> (1) Action space and SOTA baseline
>
> As discussed in more detail in our response to reviewer vxu1–Q1, the **reward formulation in RoVer only requires a metrizable continuous action space**, so it naturally extends from end-effector deltas to joint-angle actions. To concretely validate this, we perform additional real-robot experiments with **joint-space control**: we collect data via teleoperation on a dual-arm platform to fine-tune $\pi0$, and reuse the same data to train a PRM. The results are reported below:
>
> | Condition / Metric | Pour water |                 | Tidy up the desk |                 | Cook the vegetable |                 |
> | ------------------ | ---------- | --------------- | ---------------- | --------------- | ------------------ | --------------- |
> |                    | $\pi0$     | $\pi0$$\dagger$ | $\pi0$           | $\pi0$$\dagger$ | $\pi0$             | $\pi0$$\dagger$ |
> | Normal             | 50%        | 80%             | 73.3%            | 86.7%           | 30%                | 50%             |
> | Background         | 50%        | 50%             | 53.3%            | 60%             | –                  | –               |
> | New obj            | –          | –               | 46.7%            | 46.7%           | –                  | –               |
>
> Please refer to the updated draft for detailed experimental settings. These experiments show that RoVer can be **plugged into a stronger VLA backbone ($\pi0$) in joint-angle space**, providing consistent improvements on in-distribution (Normal) settings while preserving $\pi0$’s generalization performance on Background and New obj variants. A more detailed analysis of the resulting **safety behavior** (improvements on seen settings, fallback to $\pi0$ on unseen settings) is provided in #vxu1–Q1 and the **Real-robot experiments** subsection of the appendix.
>
>
>
> (2) Inference latency
>
> Thank you for the question. We now answer the question about latency.
>
> In the original Table 2, we evaluated the latency of the PRM on a V100 GPU by measuring its runtime under different numbers of candidate actions. Here we additionally report inference latency on the real-robot setup. Suppose we let the policy run $N$ times, and for each action we sample $M/N$ noise actions, leading to a total of $N{+}M$ candidates. We use $\pi0$ as the backbone and a single H100 GPU for testing. The main latency comes from $N$ forward passes of the policy and $N{+}M$ forward passes of the PRM. The results are as follows:
>
> | Total candidate | N    | M    | latency (ms) |
> | --------------- | ---- | ---- | ------------ |
> | 1               | 1    | 0    | 79.61        |
> | 3               | 3    | 0    | 203.38       |
> | 6               | 6    | 0    | 349.97       |
> | 6               | 3    | 3    | 194.65       |
> | 10              | 10   | 0    | 570.96       |
> | 10              | 5    | 5    | 309.84       |
> | 64              | 64   | 0    | 3550.96      |
> | 64              | 32   | 32   | 1878.47      |
> | 64              | 1    | 63   | 253.49       |
>
> We would like to clarify that the high-$N$ cases (e.g., $N{=}64$) are mainly relevant for **large-scale testing sweeps** on simulation benchmarks such as CALVIN, where the PRM needs to evaluate a large number of candidate actions across many sequences (e.g., GR-1 requires about 500 steps to complete one sequence consisting of 5 continuous tasks; testing 1000 sequences would require about 500,000 steps). In contrast, for a *single* real-time control step on the robot, as shown in Fig. 4 of the main text, we use at most $N{=}5$ and $M{=}5$, for which the latency remains in the hundreds of milliseconds range and is compatible with our control loop.

---

> ### Author Response · Authors · 2025-11-24
>
> > $\color{red}{W4:}$ Deeper analysis of direction prediction
>
> Thank you for the suggestion. We agree that visualizing the predicted directions is informative. However, in the CALVIN simulation environment, it is difficult to visualize in the 6D action space. This is precisely why we resorted to PushT for visualization. From the PushT visualization, we observe that the directions predicted by the PRM generally point toward the target.
>
> At the same time, we would like to point out that during training, as the direction loss decreases, the validation cosine similarity of direction predictions converges to around 0.7. This means that not all direction predictions are accurate. There are several potential directions for improvement:
> - More training data. This is the most straightforward direction, since as we state in the paper, RoVer’s PRM only uses **20%** of the CALVIN training set (randomly sampled) and already achieves the reported performance. This choice was mainly due to training-time considerations, as CALVIN is a large dataset.
> - Better loss functions or regularization. There is ample room to explore alternative designs. RoVer’s main contribution lies in proposing direction-guided training and inference. In embodied intelligence, the action space of VLAs is directly metrizable, which is a key distinction from other test-time scaling scenarios.
> - Improving direction accuracy by aggregating predictions over neighboring actions. This is also a promising direction to explore, although it would incur additional computational cost.
>
>
>
> > $\color{red}{Q1:}$ Choice of RMSE metric
>
> Thank you for the question. We chose RMSE based on a simple consideration: for a single time step, the distance between two actions can naturally be measured by RMSE. Of course, in the action-chunk scenario, other distance measures (e.g., Dynamic Time Warping) may be more appropriate. While RoVer does not directly model correlation with actual task success, under the expert-action-as-proxy assumption, the closer an action is to the expert action at each time step, the better it can reproduce the expert’s success; RoboMonkey[1] follows a similar idea. This is reasonable from an intuitive standpoint and is also supported by our current experimental results. We are very excited to explore more alternatives in future work, such as directly modeling correlation with task success, similar to recent approaches like $\pi0.6*$. We believe this direction is just beginning and there remains substantial room for exploration.
>
>
>
> > $\color{red}{Q2:}$ Inference noise-scale ablation
>
> Thank you for the question. As mentioned above in our response about real-robot experiments, we report our noise-scale ablation results below.
>
> | Noise Ablation | 1                 | 2                 | 3                     | 4                     | 5                     | Avg. Len            |
> | -------------- | ----------------- | ----------------- | --------------------- | --------------------- | --------------------- | ------------------- |
> | 0.01           | **88.3**          | **75.7**          | $\underline{65.3}$    | $\underline{55.3}$    | $\underline{45.6}$    | $\underline{3.30}$  |
> | 0.05           | 86.1              | 75.3              | **66.8**              | **56.2**              | **48.7**              | **3.33**            |
> | 0.1            | $\underline{88.0}$| 74.9              | 64.1                  | 52.8                  | 42.7                  | 3.23                |
> | 0.15           | 86.7              | 74.1              | 63.1                  | 51.6                  | 42.5                  | 3.18                |
> | 0.25           | 87.9              | 73.9              | 63.2                  | 52.5                  | 43.3                  | 3.21                |
> | 0.3            | **88.3**          | $\underline{75.4}$| 63.8                  | 52.6                  | 43.4                  | 3.24                |
>
>
> We find that an appropriate noise scale leads to clear performance improvements. Although overly large noise can perturb sampled actions, it does not significantly decrease overall performance. Instead, RoVer preserves the performance of the base policy, thanks to its safety property—the PRM tends to choose the original policy’s action when facing clearly unreasonable or unfamiliar situations, thereby guaranteeing the performance lower bound.

---

> > ### Author Response · Authors · 2025-11-24
> >
> > > $\color{red}{Q3:}$ Compatibility across base policies
> >
> > In Table 1, our experiments already include diffusion-based methods (Dita, MoDE) and a Transformer-based method (GR-1). We have not yet found two methods—one diffusion-based and one Transformer-based—with similar base performance for a more direct paired comparison. However, as RoVer is a backbone-agnostic framework, we expect its relative performance improvements to be similar across different architectures. This is also supported by the updated Table 1 in the PDF.
> >
> >
> >
> > > $\color{red}{Q4:}$ Distribution analysis and $\sigma_{base}$ choice
> >
> > We provide a more detailed explanation in the appendix of the updated PDF. All the results reported in the main text are obtained with a fixed $\sigma_{base}=0.05$, and this choice comes from our distribution analysis. In that analysis, we observe that the average distance between the actions predicted by GR-1 and the expert actions is about 0.1. This guides our choice of the range for $\sigma_{base}$. We set $\sigma_{base}=0.05$ at inference so that we can efficiently sample actions that move closer to the expert action without “overshooting” it.
> >
> >
> >
> > [1] Ji, Yuheng, et al. "Robobrain: A unified brain model for robotic manipulation from abstract to concrete." *Proceedings of the Computer Vision and Pattern Recognition Conference*. 2025.

---

### Official Review · Reviewer_nGo7 · 2025-10-29

**Soundness:** 3
**Presentation:** 3
**Contribution:** 2
**Rating:** 4
**Confidence:** 2

**Summary:**

RoVer is a test-time scaling framework designed to enhance embodied Vision-Language-Action (VLA) models without additional data collection or retraining. It introduces a lightweight Process Reward Model (PRM) that both scores candidate robot actions and predicts refinement directions in the action space, at inference time. By generating and verifying many candidate actions, RoVer stabilizes long-horizon performance and unlocks latent capabilities in existing policies. Experiments in simulation and real-world manipulation tasks demonstrate consistent performance improvements in both simulation and the real-world setup.

**Strengths:**

- RoVer introduces a plug-and-play test-time scaling framework that complements traditional training-time scaling, providing a new horizontal improvement direction for existing VLA policies without modifying their architectures or weights.

- The method makes efficient use of perception caching, enabling the evaluation of a large number of candidate actions with minimal additional compute.

- RoVer directly addresses a key bottleneck in embodied AI, the high cost of collecting and annotating robotic data, by reallocating compute to inference-time verification and refinement, achieving meaningful performance gains without expanding the dataset with annotation.

**Weaknesses:**

- While the high cost of collecting robotic data is a clear motivation for test-time scaling, the claim that *"VLA success rates fluctuate due to stochastic decoding and manipulation brittleness, especially for long-horizon tasks"* remains underspecified. Please clarify how this observation motivates reallocating computation from training to inference.

- Some important baselines are missing. In particular, evaluation on stronger or more recent state-of-the-art VLA backbones (e.g., Pi-zero[1]) would better demonstrate generality and competitiveness.

- It is unclear whether $R_\phi$(h, a) outputs only a scalar reward or both reward and direction (lines 171 and 279).

- Figure 1 over-emphasizes the base inference module rather than RoVer’s novel components, and Figure 2 is overly dense and hard to interpret. Simplifying visuals to highlight the contribution would improve accessibility.


---
[1] Black, Kevin, et al. "$\pi_0 $: A Vision-Language-Action Flow Model for General Robot Control." RSS (2025).

**Questions:**

- Why is the way to compute adaptive noise scale necessary, rather than sampling noisy expert action $a_{en}$ around $a_e$ and distinguishing better and worse by computing distance to expert actions? Although the authors mention it yielded poor performance, what is the main reason that makes these two significantly different?

- RoVer's benefit appears dependent on the PRM being trained from small perturbations around expert actions. In cases where the base policy deviates significantly from the expert distribution (e.g., early failures in long-horizon tasks), does PRM guidance remain reliable? Do the authors have relevant observations in the experiments?

- Is a separate PRM trained for each manipulation task, or can a single model generalize across multiple tasks? For diverse tasks with different action distributions, how sensitive is the method to the chosen Gaussian noise scale during training? Results on multi-task PRM or noise-scale ablations would clarify scalability.

---

> ### Author Response · Authors · 2025-11-24
>
> > $\color{red}{W1:}$ Motivation for test-time scaling
>
> Thank you for the question. Here we provide a more detailed explanation of the claim that you highlighted and how it motivates **reallocating computation from training to inference**. Current VLA backbones are typically based on either autoregressive Transformer architectures or diffusion-based architectures. In both cases, inference involves probabilistic sampling (e.g., decoding temperature, diffusion noise), so the **same trained policy can produce different action sequences for the same task**. Test-time scaling in LLMs leverages exactly this property[1], where an external PRM selects among multiple trajectories generated by a fixed base model.
>
> In embodied manipulation, this stochasticity interacts with **contact-rich, brittle dynamics**: for the same goal, we often observe that a frozen policy sometimes succeeds and sometimes fails, even though the underlying observations and prompts are similar. The successful rollouts demonstrate that the policy already **contains the capability to solve the task**, while failures are largely due to random deviations in the sampled actions that get amplified over long horizons. This suggests that simply collecting more data and retraining the base policy is not the only—or even the most direct—way to improve performance; instead, it is natural to spend additional **test-time compute** to (i) generate multiple candidate actions from the existing policy and (ii) use a verifier to pick the ones that are more likely to lead to success.
>
> RoVer operationalizes this idea for VLAs: a **lightweight PRM with dense process rewards and action-space directions** evaluates and refines candidate actions at inference, turning extra compute into reduced variance and improved reliability without modifying the base model’s weights. Empirically, as shown in the updated Table 1 on the CALVIN long-horizon benchmark, RoVer yields **larger gains on later tasks in the ABC$\rightarrow$D sequence**, where error accumulation is more severe. This pattern is consistent with our motivation that test-time verification is particularly valuable for mitigating stochastic failures in long-horizon manipulation.
>
>
>
> > $\color{red}{W2:}$ Missing stronger VLA baselines
>
> Thank you for the comment. To address your concern about **stronger or more recent VLA backbones**, we additionally evaluate RoVer on $\pi0$—a recent state-of-the-art generalist VLA policy—in a real-robot setting. On our real-robot platform, we collect data via teleoperation to fine-tune $\pi0$, and we use the same real-robot data to train a PRM. We report the results below.
>
> | Condition / Metric | Pour water |                 | Tidy up the desk |                 | Cook the vegetable |                 |
> | ------------------ | ---------- | --------------- | ---------------- | --------------- | ------------------ | --------------- |
> |                    | $\pi0$     | $\pi0$$\dagger$ | $\pi0$           | $\pi0$$\dagger$ | $\pi0$             | $\pi0$$\dagger$ |
> | Normal             | 50%       | 80%            | 73.3%            | 86.7%           | 30%             | 50%            |
> | Background         | 50%       | 50%            | 53.3%            | 60%            | –                  | –               |
> | New obj            | –         | –              | 46.7%            | 46.7%          | –                  | –               |
>
> Please refer to the updated draft for full experimental details. In these real-robot experiments based on $\pi0$, we observe that RoVer provides **consistent improvements** over the fine-tuned $\pi0$ on in-distribution (Normal) settings across all three tasks (e.g., Normal success rates increase from 50% to 80% on "Pour water" and from 30% to 50% on "Cook the vegetable"), while preserving $\pi0$’s **generalization** performance on Background and New obj variants.
>
> We analyze this behavior in more depth—including the **safety** phenomenon where PRM improves over $\pi0$ on seen settings but falls back to $\pi0$ on unseen scenes—in our response to reviewer #vxu1–Q1 and in the **Real-robot experiments** and **Noise-scale ablation** subsections of the appendix. Here, these results primarily serve to show that RoVer remains effective when plugged into a stronger, modern VLA backbone such as $\pi0$.

---

> ### Author Response · Authors · 2025-11-24
>
> > $\color{red}{W3:}$ Clarify PRM outputs
>
> Thank you for the question. We now clarify the outputs of $R_{\phi}(h,a)$. For all instances of $R_{\phi}(h,a)$ mentioned in the paper, the PRM predicts **both (i) a scalar process reward and (ii) an action-space direction**. During training, we use $L_{\text{rew}}$ to train the reward head and $L_{\text{dir}}$ to train the direction head. At inference time, we first let the policy model generate multiple candidate policy actions, and then use the PRM to obtain the reward and direction for each candidate action. Next, for each policy action we sample noise actions along the corresponding direction and evaluate their rewards with the PRM. Finally, we rank all rewards and execute the action with the highest reward. We have added a more detailed algorithmic description in the appendix of the updated PDF, and we kindly invite you to refer to it.
>
>
>
> > $\color{red}{W4:}$ Clarity of figures and visuals
>
> Thank you very much for the suggestion. Due to time constraints, we prioritized adding a large number of experiments in the current submission. We plan to refine and update the figures in subsequent revisions.
>
>
>
> > $\color{red}{Q1:}$ Adaptive noise scale design
>
> Thank you for the careful reading. Here we clarify the intended meaning in the paper.
>
> By adaptive noise scale, we refer to the mechanism used to distinguish between $a_{better}$ and $a_{worse}$. Concretely, in the scheme you mentioned (which is also our initial attempt), for an expert action $a_e$ we sample two actions around it using a fixed noise scale $\sigma_{base}$, and then compute their distances via RMSE to obtain $a_{better}$ and $a_{worse}$. This leads the PRM to focus on distinguishing actions centered around the expert action $a_e$. Under a fixed $\sigma_{base}$, these two actions are usually easy to distinguish during training but less effective at test time when facing more complex scenarios.
>
> In our current method, we also sample $a_{anc}$ from $a_e$ using a fixed noise scale $\sigma_{base}$. Although $\sigma_{base}$ is fixed, the actual distance between $a_e$ and the sampled $a_{anc}$ is random. We then treat the distance between $a_e$ and $a_{anc}$ as a dynamic $\sigma_{adapt}$, and sample $a_{better}$ and $a_{worse}$ within a neighborhood around $a_{anc}$ determined by $\sigma_{adapt}$. Since $\sigma_{adapt}$ is dynamic, when $a_{anc}$ is close to $a_e$, the neighborhood induced by $\sigma_{adapt}$ is small, allowing the PRM to learn to distinguish finer differences between nearby samples; when $a_{anc}$ is further away, the neighborhood is expanded accordingly. In our experiments (see the **Noise-scale ablation and training diagnostics in the appendix**), we find that the initial fixed-$\sigma_{base}$ scheme leads to relatively easy training pairs but weaker ranking quality at test time, whereas the anchor-centered adaptive scheme better matches the distribution of test-time candidates and yields stronger long-horizon performance.
>
> We have further elaborated this mechanism in the algorithmic description in the appendix of the updated draft.
>
>
>
> > $\color{red}{Q2:}$ PRM reliability in long-horizon failures
>
> Thank you for the question. On the one hand, the notion of "small perturbation" is relative. With the adaptive noise strategy described above, the random sampling scheme allows RoVer’s PRM to observe **noise actions at a range of distances from expert actions** during training, including actions that are relatively far as well as those that are close. This is also why we choose three policy models with different performance levels while using a single PRM for all of them: even for relatively weak base policies such as GR-1, RoVer still yields significant long-horizon gains in Table 1 in updated draft, indicating that the learned PRM is not limited to extremely small perturbations.
>
> For samples that **deviate significantly from the expert distribution**, the predicted direction plays a crucial role. Starting from the policy action, the PRM suggests an action-space direction that points toward improved behavior, and we sample along this direction, gradually pulling the policy’s actions toward the neighborhood of the expert actions. At the same time, as discussed in our safety analysis (#vxu1–Q1 and the Noise-scale ablation), when all candidates are poor (e.g., heavily perturbed by large noise), the PRM tends to avoid clearly unreasonable actions and falls back to the base policy’s proposal, so RoVer does not catastrophically degrade performance even when the base policy is temporarily far from the expert distribution.

---

> ### Author Response · Authors · 2025-11-24
>
> > $\color{red}{Q3:}$ PRM training recipe and noise ablation
>
> *Q3.1* Training recipe of PRM
>
> We train the PRM in a **multi-task** manner. That is, as shown in Table 1, on the CALVIN dataset **we use the $\underline{same}$ PRM across all base models and all evaluation tasks**, without any task-specific modifications. This design aligns with RoVer’s plug-and-play goal and empirically demonstrates that a single PRM can generalize across diverse tasks and policies, rather than needing a separate verifier per task.
>
> *Q3.2* Regarding your question about **noise-scale ablations and sensitivity**, we report our results below.
>
> | Noise Ablation | 1                 | 2                 | 3                     | 4                     | 5                     | Avg. Len            |
> | -------------- | ----------------- | ----------------- | --------------------- | --------------------- | --------------------- | ------------------- |
> | 0.01           | **88.3**          | **75.7**          | $\underline{65.3}$    | $\underline{55.3}$    | $\underline{45.6}$    | $\underline{3.30}$  |
> | 0.05           | 86.1              | 75.3              | **66.8**              | **56.2**              | **48.7**              | **3.33**            |
> | 0.1            | $\underline{88.0}$| 74.9              | 64.1                  | 52.8                  | 42.7                  | 3.23                |
> | 0.15           | 86.7              | 74.1              | 63.1                  | 51.6                  | 42.5                  | 3.18                |
> | 0.25           | 87.9              | 73.9              | 63.2                  | 52.5                  | 43.3                  | 3.21                |
> | 0.3            | **88.3**          | $\underline{75.4}$| 63.8                  | 52.6                  | 43.4                  | 3.24                |
>
>
> We find that an **appropriate inference noise scale** leads to clear performance improvements: around $\sigma\\in\[0.01,0.3]$, performance is consistently strong, with the best Avg. Len achieved at $\sigma{=}0.05$, which we adopt in our main experiments. Importantly, the curve is **fairly flat in this moderate range**, indicating that RoVer is not overly sensitive to the exact choice of noise scale. When the noise is too large, it does perturb the sampled actions; however, this does not significantly degrade overall performance. Instead, RoVer maintains a similar performance level, thanks to its **safety** property (also mentioned in our response about real-robot experiments above and updated draft)—when facing obviously unreasonable or unfamiliar candidates, the PRM tends to prefer the original policy action, thereby preserving the performance lower bound of the RoVer framework. We provide additional discussion of these trends in the **Noise-scale ablation** subsection of the appendix.
>
>
>
> [1] Snell, Charlie, et al. "Scaling llm test-time compute optimally can be more effective than scaling model parameters." *arXiv preprint arXiv:2408.03314* (2024).

---

### Official Review · Reviewer_vxu1 · 2025-10-31

**Soundness:** 2
**Presentation:** 3
**Contribution:** 3
**Rating:** 4
**Confidence:** 4

**Summary:**

This paper proposes an inference expansion framework named RoVer, which aims to enhance the performance of existing Vision-Language-Action (VLA) models without modifying their architectures or weights. RoVer adopts a Robotic Process Reward Model (PRM) as the runtime validator. During the inference phase, RoVer synchronously generates multiple candidate actions from the base policy, expands them along the directions predicted by the PRM, and selects the optimal action for execution through PRM scoring. Experiments demonstrate that RoVer consistently improves success rates across various manipulation tasks, providing a runtime expansion solution for VLA models that requires no additional training overhead.

**Strengths:**

1. The method proposed in the paper is highly lightweight and exhibits strong scalability.
2. The writing follows a clear and rigorous logical flow, making it easy to understand.
3. The experimental section is extremely detailed and comprehensive.

**Weaknesses:**

1. **The comparison with crucial related work is missing**: *Nakamoto, M., Mees, O., Kumar, A., & Levine, S. (2024). Steering your generalists: Improving robotic foundation models via value guidance. arXiv preprint arXiv:2410.13816.*
This paper also selects the action with the maximum Value during the inference stage, which is very similar to the core idea of the RoVer. Please make a detailed comparison of the similarities and differences between RoVer and V - GPS.

2. **The selection of $a_{better}$ may not be better**: The anchor action $a_{anc}$ is obtained by adding noise to the expert action $a_e$. Based on the anchor action $a_{anc}$, the action closer to $a_e$ is called $a_{better}$.
Suppose a successful trajectory is $a_1$, $a_2$, $a_3$ until the task is completed. If $a_2$ is taken as the current expert action, it is possible that $a_{anc}$ obtained after adding noise is closer to $a_3$. At this time, $a_{better}$ obtained from $a_2$ close to the expert action may be worse than the expert action instead.
What impact will this situation have on the learning of rewards?

**Questions:**

1. Is the current reward function compatible with the VLA algorithm for joint angle control?
2. How should the hyperparameters in Equation 14 be selected, and what impact do they have on the corresponding performance?

---

> ### Author Response · Authors · 2025-11-24
>
> > $\color{red}{W1:}$ Comparison with V-GPS
>
> Thank you very much for raising this point. We have already provided a detailed comparison with V-GPS in our response to reviewer UurV–Q2, including both overall results and an analysis of reward hacking behavior. Here we briefly summarize the empirical numbers for completeness (please refer to the updated Table 1 in the PDF for the full table and to UurV–Q2 for a more in-depth discussion).
>
> | Method        | 1     | 2     | 3     | 4     | 5     | Avg. Len |
> | ------------- | ----- | ----- | ----- | ----- | ----- | -------- |
> | GR-1 w/ V-GPS | 86.4  | 73.0  | 62.3  | 50.8  | 40.5  | 3.13     |
> | $\Delta$      | $\color{green}{+1.3}$ | $\color{red}{-0.7}$ | $\color{red}{-0.9}$ | $\color{red}{-2.9}$ | $\color{red}{-2.9}$ | $\color{red}{-0.6}$ |
> | Dita          | 93.2  | 83.2  | 72.8  | 64.4  | 53.2  | 3.67     |
> | $\Delta$      | $\color{red}{-1.3}$ | $\color{green}{+0.7}$ | 0 | $\color{green}{+3.1}$ | $\color{green}{+3.2}$ | $\color{green}{+0.06}$ |
> | MoDE          | 96.4  | 89.9  | 82.1  | 72.8  | 65.5  | 4.07     |
> | $\Delta$      | $\color{green}{+0.2}$ | $\color{green}{+1.0}$ | $\color{green}{+1.0}$ | $\color{green}{+1.0}$ | $\color{green}{+2.0}$ | $\color{green}{+0.06}$ |
>
> As summarized in UurV–Q2, these results show that V-GPS can bring noticeable gains on stronger backbones such as Dita and MoDE, but its impact is not uniformly positive (e.g., GR-1 sees degraded long-horizon performance and Avg. Len), and the GR-1 N+M ablation reveals reward hacking as the number of candidates increases. RoVer, by contrast, uses dense process rewards and direction-guided sampling, and under comparable candidate budgets yields more stable improvements across different policies on CALVIN.
>
> Specifically, V-GPS uses sparse rewards over the last $H$ steps and treats the offline RL value function as a test-time critic. This leads to several issues:
> 1. As discussed by ReinBoT[2], **sparse rewards inherently suffer from credit assignment problems**, especially for long-horizon, multi-subgoal, or high-precision manipulation tasks that are common in visual robotics. Sparse rewards only provide a binary signal at the final success, and cannot reflect quality differences among intermediate steps or capture many critical aspects of manipulation (e.g., pose adjustment before grasping, smoothness of motion, or whether intermediate subgoals are completed). Hence, value functions based on sparse rewards often struggle to distinguish “almost correct actions” from truly high-quality actions and are prone to unstable or over-optimistic estimates.
> 2. In contrast, RoVer directly focuses on the **relative ordering among candidate actions**, and learns finer-grained process rewards via pairwise preference. This allows RoVer to capture subtle differences in action quality that sparse rewards cannot reflect, such as the timing of gripper closure or small geometric deviations in the approach trajectory, making it more suitable for fine-grained visual manipulation tasks.
> 3. Furthermore, **even if we do not use direction-guided sampling (DG) at inference**, DG in the training phase significantly improves PRM training efficiency(please see Fig. 7 in the appendix). The predicted improvement direction provides more structured and dense supervision, helping the model converge faster. We elaborate on this in more detail in our response to Q2: direction supervision lets the PRM know not only "which action is better", but also "in which direction it should move", which greatly enhances its learning ability—a benefit that a sparse-reward-based critic cannot provide during training.

---

> ### Author Response · Authors · 2025-11-24
>
> > $\color{red}{W2:}$ Definition of $a_{better}$ from anchor
>
> We very much appreciate your insightful comment! At its core, this question is closely related to the **PRM sequence-modeling** issue that we discussed in our response to reviewer #UurV. In the current work, we deliberately adopt a **per-step proxy scheme**: the PRM is trained to compare actions with respect to the expert at the current time step, while long-horizon behavior emerges implicitly from repeatedly applying this preference at test time. This design matches our main focus on **direction-based test-time scaling**, and follows RoboMonkey[1], where a similar per-step expert-proximity proxy is shown to be effective in practice.
>
> The example you mentioned—where an anchor sampled from $a_2$ may lie closer to a future expert action such as $a_3$ or $a_5$—highlights exactly the kind of temporal structure that a more explicit sequence model could exploit: **such cases need not be treated as "wrong"**, but rather as moving along the demonstrated trajectory toward a later subgoal. Our current per-step formulation does not yet distinguish these situations and instead treats them purely through the local distance to the current expert; nevertheless, the overall results in Table 1 suggest that this simple proxy already works well empirically.
>
> That said, we agree that **explicitly incorporating future expert actions into PRM training** is a promising way to further improve efficiency. As you suggested, a PRM that reasons over short action subsequences could intentionally prefer an action closer to $a_5$ rather than the original $a_2$, effectively skipping over $a_2, a_3, a_4$ and potentially reducing the number of steps required for the policy to complete a task. We view this sequence-modeling extension as complementary to RoVer’s **core contribution on direction-guided test-time scaling**, and plan to explore it in future work.

---

> ### Author Response · Authors · 2025-11-24
>
> > $\color{red}{Q1:}$ Compatibility with joint-angle VLA control
>
> Thank you for the professional comments. The **reward function in RoVer only assumes a metrizable continuous action space**: at training time, we measure RMSE between candidate actions and expert actions in the underlying action coordinates. For joint-angle control, we use the same definition with joint angles as the action representation, so **no change to the reward formulation or training objective is required**.
>
> On our real-world platform, we collected **joint-angle-space actions via teleoperation** and fine-tuned $\pi0$ on this data, and simultaneously trained a PRM using the same real-robot data. The experimental results confirm that RoVer is compatible with joint-space control, since the core idea of RoVer is to encourage the final executed action—regardless of whether it is in end-effector or joint coordinates—to be close to the expert action in the corresponding action metric.
>
> | Condition / Metric | Pour water |                 | Tidy up the desk |                 | Cook the vegetable |                 |
> | ------------------ | ---------- | --------------- | ---------------- | --------------- | ------------------ | --------------- |
> |                    | $\pi0$     | $\pi0$$\dagger$ | $\pi0$           | $\pi0$$\dagger$ | $\pi0$             | $\pi0$$\dagger$ |
> | Normal             | 50%       | 80%            | 73.3%            | 86.7%           | 30%             | 50%            |
> | Background         | 50%       | 50%            | 53.3%             | 60%            | –                  | –               |
> | New obj            | –          | –               | 46.7%             | 46.7%            | –                  | –               |
>
> Please refer to the updated PDF for detailed experimental settings. In the real-robot experiments based on $\pi0$, we observe the following:
>
> 1. When the test distribution matches the training distribution, RoVer shows clear effectiveness.
> 2. Leveraging the large-scale pre-training of $\pi0$, we also evaluate the model’s **generalization** ability.
>
> From this, we observe an additional strength of RoVer—we refer to it as **safety**: suppose the first action predicted by $\pi0$ is denoted as the base action. In our experiments, under the Normal setting of each task, RoVer frequently selects alternative actions different from the base action to improve success rates. However, under variant settings such as Background or New obj, the new scenes and target objects do not appear in the PRM’s training data (since the PRM is not yet pre-trained at scale). In such cases, the PRM always assigns the highest reward to the base action. As a result, for tasks that require generalization to new backgrounds or objects, $\pi0$ still performs well thanks to its large-scale pre-training, while the PRM—without large-scale pre-training—**avoids being over-confident on unfamiliar situations and instead firmly selects the base action as the final execution**. This ensures that, under distribution shift, the PRM trained within the RoVer framework preserves the performance lower bound of the overall system.
>
> Consistently, in the **noise-scale ablation on inference reported for reviewer #nGo7-Q3 and #xijL-Q2**, we observe the same safety behavior: when the noise scale becomes very large and sampled actions are heavily perturbed, the PRM rarely chooses obviously unreasonable candidates and instead falls back to the base policy’s action, so the overall performance does not degrade below the base policy. We refer the reviewer to the **Noise-scale ablation** subsection in the appendix for detailed quantitative results.

---

> ### Author Response · Authors · 2025-11-25
>
> > $\color{red}{Q2:}$ Loss hyperparameters in Eq.14
>
> Thank you for the detailed reading. In Eq. (14), since the reward and direction predictions correspond to different sub-tasks, we set their weights to be equal (both 1) in our experiments. Furthermore, we conducted ablations to clarify our contribution. We highlight that **even when direction-guided sampling (DG) is not used at inference time**, adding DG as an auxiliary signal during training can still improve PRM performance. In the appendix of the paper, we include plots of our validation metrics during training; since we cannot include figures here, please refer to the paper for details.
>
> Specifically, we compare two training settings—training with DG and without DG (direction loss weight = 0)—and in both cases use only unguided sampling at inference (i.e., no DG expansion). The following table reports GR-1 performance on CALVIN ABC$\rightarrow$D under different candidate configurations $N{+}M$. We can see that in most configurations, adding DG supervision improves the Avg. Len and SR@k for long horizons, particularly when the number of candidates is larger. These results have been incorporated into the appendix in a more formal format.
>
> Overall, this ablation supports our choice of **balanced weights in Eq. (14)** and confirms that **training time direction supervision itself is a key contributor to RoVer’s gains**, independent of whether DG is used at inference time.
>
> | GR-1 N+M     | 1    | 2    | 3    | 4    | 5    | Avg. Len |
> | ------------ | ---- | ---- | ---- | ---- | ---- | -------- |
> | 3+0 (w/o DG) | **87.5** | 74.0 | **64.2** | **54.5** | **44.7** | **3.249** |
> | 3+0 (w/ DG)  | 87.3 | **75.0** | **64.2** | 54.1 | 44.3 | **3.249** |
> | ---          | ---  | ---  | ---  | ---  | ---  | ---      |
> | 5+0 (w/o DG) | **87.3** | 74.4 | 63.3 | 53.2 | 43.1 | 3.213 |
> | 5+0 (w/ DG)  | 86.8 | **74.6** | **64.1** | **54.3** | **45.3** | **3.251** |
> | ---          | ---  | ---  | ---  | ---  | ---  | ---      |
> | 1+5 (w/o DG) | **87.5** | **74.4** | 63.7 | 53.1 | 42.9 | 3.216 |
> | 1+5 (w/ DG)  | 86.6 | 74.3 | **64.2** | **54.7** | **46.1** | **3.259** |
> | ---          | ---  | ---  | ---  | ---  | ---  | ---      |
> | 3+3 (w/o DG) | 87.3 | 74.3 | 63.1 | 52.8 | 42.1 | 3.196 |
> | 3+3 (w/ DG)  | **87.4** | **74.5** | **64.5** | **54.4** | **45.3** | **3.261** |
> | ---          | ---  | ---  | ---  | ---  | ---  | ---      |
> | 5+5 (w/o DG) | **87.7** | 73.6 | 62.8 | 52.7 | 43.4 | 3.202 |
> | 5+5 (w/ DG)  | 86.5 | **73.8** | **64.5** | **53.8** | **44.9** | **3.235** |

---

### Official Review · Reviewer_UurV · 2025-11-01

**Soundness:** 2
**Presentation:** 3
**Contribution:** 2
**Rating:** 4
**Confidence:** 4

**Summary:**

This paper introduces RoVer, a test-time scaling framework for VLA models that enhances robotic policy performance without modifying the base model's architecture or weights. RoVer employs a lightweight PRM to score candidate actions and predict refinement directions in the action space. Experiments on the CALVIN benchmark and real-robot tasks demonstrate consistent improvements across multiple base policies (GR-1, Dita, MoDE, and Diffusion Policy), with significant gains in success rates and inference efficiency.

**Strengths:**

1. Rigorous experiments across simulation and real robots, with strong empirical results;
2. Offers a practical and efficient alternative to costly training-time scaling, with broad applicability to existing policies and VLA models.

**Weaknesses:**

1. The method is less effective for chunk-based policies like MoDE due to the step-chunk mismatch;
2. The PRM relies on expert action proximity as a supervision proxy, which may not always align with task success in more complex or long-horizon settings.

**Questions:**

1. The authors themselves also mentioned the limitation of the mismatch between chunk-based policy inference and PRM inference. Could the direction-guided sampling be extended to handle multi-step or hierarchical actions to better support chunk-based policies?
2. How does the performance of the authors' proposed method compare to the RL-based approach [1] that uses a trained value function to guide policy sampling?

[1] Nakamoto, et al. Steering Your Generalists: Improving Robotic Foundation Models via Value Guidance. CoRL, 2024.

---

> ### Author Response · Authors · 2025-11-24
>
> > $\color{red}{W1\\&Q1:}$ Step–chunk mismatch in chunk policies
>
> Thank you for the careful review.
>
> - **Scope and core focus.** As we explicitly acknowledge in the main paper, the current RoVer framework indeed has a mismatch when applied to action‑chunk policies. However, in contrast to prior work like V-GPS[3] and our concurrent work RoboMonkey[1], the **core contribution** of RoVer is to propose an embodied test‑time scaling framework that leverages the **metrizable action space** of VLA policies: by modeling distances and directions in action space, we improve both PRM training and inference. Although this mismatch lies outside the discussion framework we intend for RoVer, we highlight it because we see it as a worthwhile direction for future work.
>
> - **How we apply RoVer to MoDE.** When applying RoVer’s sampling to MoDE, we evaluate the PRM **only on the first timestep of each action chunk** and select the chunk whose first action receives the highest reward. In our experiments, extending the PRM evaluation to the entire chunk does not bring notable additional gains.
>
> | strategy                          | 1        | 2        | 3        | 4        | 5        | Avg. Len |
> | --------------------------------- | -------- | -------- | -------- | -------- | -------- | -------- |
> | first-step reward (MoDE$\dagger$) | **97.1** | **90.9** | **82.5** | **74.9** | **66.6** | **4.12** |
> | whole-chunk reward                | 96.4     | 88.8     | 80.4     | 72.8     | 63.4     | 4.02     |
>
> - **Empirical behavior.** Although this mismatch exists, Fig. 3 (left) shows that RoVer’s current selection strategy still yields **stable improvements on MoDE** under our experimental settings, even if the overall gain is modest.
>
>
> > $\color{red}{W2:}$ Expert proximity as supervision proxy
>
> Thank you for this helpful comment.
>
> - **Conceptual relation to future-return supervision.** The suggestion of incorporating supervision from task-level future rewards can be viewed as a natural extension of our current design: in addition to modeling action chunks, one could directly train the PRM to predict long-horizon returns (e.g., discounted returns in RL or RTG in ReinBoT[2]). We agree this is an exciting direction for future work.
>
> - **Orthogonality to RoVer’s main focus.** At the same time, we would like to stress that this extension is **largely orthogonal to the main focus of RoVer**: as discussed in W1\&Q1, the **core contribution of RoVer lies in how we exploit action-space directions both during training (via the direction loss) and during inference (via direction-guided sampling)**, leveraging the metrizable continuous action space of VLAs to realize a form of test-time scaling that is qualitatively _different from existing TTS approaches in other domains_.
>
> - **Why expert proximity is a reasonable proxy.** In the present work, RoVer deliberately uses expert action proximity as a dense per-step process reward. On CALVIN, demonstrations are dominated by successful trajectories, so staying close to the expert sequence keeps the policy on a known-successful manifold, while drifting away tends to accumulate errors in long-horizon manipulation. From this perspective, expert proximity provides a low-variance surrogate for outcome rewards that is well aligned with task success in our setting, and a similar idea is also adopted in RoboMonkey[1].
>
> - **Empirical support.** RoVer trained with this proxy consistently improves downstream performance. In the updated draft of Table 1 we add rows reporting absolute gains, and the excerpt below shows that on CALVIN—a widely used long-horizon benchmark—the improvements are particularly pronounced on later steps in the ABC$\rightarrow$D sequence. This suggests that the per-step proximity proxy effectively propagates into better long-horizon success under our experimental regime, while more explicit return modeling (as you propose) is a promising next step beyond the current work.
>
>
> | Method | 1    | 2    | 3    | 4    | 5    | Avg. Len |
> | ------ | ---- | ---- | ---- | ---- | ---- | -------- |
> | GR-1   | $\color{green}{+1.0}$ | $\color{green}{+1.6}$ | $\color{green}{+3.6}$ | $\color{green}{+2.5}$ | $\color{green}{+5.3}$ | $\color{green}{+0.14}$ |
> | Dita   | $\color{green}{+0.3}$ | $\color{green}{+0.7}$ | $\color{green}{+4.0}$ | $\color{green}{+8.7}$ | $\color{green}{+9.2}$ | $\color{green}{+0.23}$ |
> | MoDE   | $\color{green}{+0.9}$ | $\color{green}{+2.0}$ | $\color{green}{+1.4}$ | $\color{green}{+3.1}$ | $\color{green}{+3.1}$ | $\color{green}{+0.11}$ |
>
> As can be seen from the V-GPS comparison in Q2, V-GPS provides smaller improvements than RoVer overall and exhibits more severe degradation on a weaker baseline such as GR-1, which is consistent with the **reward hacking** phenomenon reported in RoboMonkey[1].

---

> ### Author Response · Authors · 2025-11-24
>
> > $\color{red}{Q2:}$ Comparison with V-GPS
>
> Thank you very much for pointing this out. We have reproduced and evaluated V-GPS[3] on the CALVIN dataset, and we report our results below.
>
> The following table shows the performance of V-GPS[3] on different base models in the CALVIN simulation environment (please refer to the updated Table 1 in the PDF for the full table).
>
> | Method        | 1     | 2     | 3     | 4     | 5     | Avg. Len |
> | ------------- | ----- | ----- | ----- | ----- | ----- | -------- |
> | GR-1 $\ddagger$ | 86.4  | 73.0  | 62.3  | 50.8  | 40.5  | 3.13     |
> | $\Delta$      | $\color{green}{+1.3}$ | $\color{red}{-0.7}$ | $\color{red}{-0.9}$ | $\color{red}{-2.9}$ | $\color{red}{-2.9}$ | $\color{red}{-0.6}$ |
> | Dita $\ddagger$         | 93.2  | 83.2  | 72.8  | 64.4  | 53.2  | 3.67     |
> | $\Delta$      | $\color{red}{-1.3}$ | $\color{green}{+0.7}$ | 0 | $\color{green}{+3.1}$ | $\color{green}{+3.2}$ | $\color{green}{+0.06}$ |
> | MoDE  $\ddagger$        | 96.4  | 89.9  | 82.1  | 72.8  | 65.5  | 4.07     |
> | $\Delta$      | $\color{green}{+0.2}$ | $\color{green}{+1.0}$ | $\color{green}{+1.0}$ | $\color{green}{+1.0}$ | $\color{green}{+2.0}$ | $\color{green}{+0.06}$ |
>
>
> In addition, based on GR-1 we evaluated V-GPS[3] under different numbers of candidate actions:
>
> | GR-1 N+M | 1                     | 2                    | 3                        | 4                        | 5                        | Avg. Len                    |
> |---------|------------------------|----------------------|--------------------------|--------------------------|--------------------------|------------------------------|
> | 3+0     | **86.5**               | 71.7                | $\underline{61.7}$       | **51.7**                 | $\underline{40.6}$       | $\underline{3.122}$         |
> | 5+0     | **86.5**               | $\underline{71.9}$   | 60.8                     | 50.4                     | **41.5**                 | 3.111                       |
> | 3+3     | $\underline{86.4}$     | **73.0**            | **62.3**                 | $\underline{50.8}$       | 40.5                     | **3.13**                    |
> | 1+5     | 78.8                  | 56.6                | 39.3                     | 29.1                     | 22.4                     | 2.262                       |
> | 5+5     | 78.5                  | 55.2                | 38.8                     | 27.1                     | 18.6                     | 2.182                       |
>
>
> - **Behavior of V-GPS.** V-GPS yields clear gains on stronger backbones such as Dita and MoDE, but performs **poorly on GR-1**, where longer-horizon metrics and Avg. Len are degraded. The GR-1 N+M ablation further shows that **increasing the number of candidate actions leads to clear performance drops**, consistent with the "reward hacking" phenomenon reported in RoboMonkey[1]. For test-time scaling, this contrast is important: on strong base policies, performance gains are relatively easier because the base already provides strong performance and a high performance floor, whereas on weaker baselines, **stable improvements** are a better indicator of the verifier’s ability to make reliable judgments.
>
> - **Why RoVer is designed differently.** In RoVer, instead of relying on a sparse value critic, we train a lightweight PRM on **dense per-step process rewards** and additionally supervise an **action-space direction head**, then use direction-guided expansion at test time under a fixed candidate budget. Under this design, RoVer achieves more stable gains on CALVIN and, as discussed in W1\&Q1, exhibits a **monotonic or at least consistently improving trend** as we increase the number of candidates.
>
>
>
> [1] Kwok, Jacky, et al. "RoboMonkey: Scaling Test-Time Sampling and Verification for Vision-Language-Action Models." CoRL, 2025
>
> [2] Zhang, Hongyin, et al. "ReinboT: Amplifying Robot Visual-Language Manipulation with Reinforcement Learning." ICML, 2025
>
> [3] Nakamoto, et al. Steering Your Generalists: Improving Robotic Foundation Models via Value Guidance. CoRL, 2024.

---

### Author Response · Authors · 2025-11-27

Dear Reviewer,

I hope this message finds you well. As the discussion period is nearing its end with less than a week remaining, we wanted to check in to ensure that we have fully addressed your concerns. If there are any additional points or clarifications you would like us to consider, please feel free to let us know. Your feedback has been invaluable, and we are eager to address any remaining issues to further improve our work.

Thank you again for your time and effort in reviewing our paper.

---

### Author Response · Authors · 2025-12-01
**Summary of rebuttal**

**Dear Program Chairs, Senior Area Chairs, Area Chairs, and Reviewers,**

We sincerely appreciate your time and extra effort in evaluating our submission, especially under the challenging circumstances of this review cycle. Since detailed back-and-forth discussion has not been possible and we have not yet had the opportunity to receive direct feedback from the reviewers, we briefly summarize how our rebuttal addresses the main concerns:

- **Mismatch with action-chunk policies (UurV–W1&Q1).** The reviewers’ observation on this point is something we have already proactively acknowledged in the Conclusion section in our draft, as we view it as a worthwhile direction for future exploration. We also emphasize that RoVer’s **core contribution** is an embodied test-time scaling framework built around metrizable action spaces and direction-guided sampling, and we explicitly point to chunk-level PRM as a natural direction for future work.
- **Expert action as proxy and alignment with task success (UurV–W2, xijL-Q1).** We explain why expert proximity is a reasonable dense per-step process reward and provide empirical evidence that this proxy yields larger gains on later tasks in ABC→D.
- **Comparison with V-GPS (UurV–Q2, vxu1–W1).** We add a thorough V-GPS baseline on CALVIN and show that while V-GPS can improve strong backbones (Dita, MoDE), it degrades GR-1 and suffers from reward hacking as the number of candidates grows. In contrast, RoVer’s process rewards and direction-guided sampling yield more stable gains and a monotonic (or consistently improving) trend as candidate counts increase within our evaluated range. [section 4.1, Table 1]
- **Compatibility with joint-angle control and stronger VLA backbones (vxu1–Q1, nGo7–W2, xijL–W2&W3).** We clarify that RoVer’s reward formulation only assumes a metrizable continuous action space, making it naturally applicable to joint-angle actions. We further add real-robot experiments where RoVer is plugged into **$\pi0$** in joint space: RoVer consistently improves in-distribution performance while preserving $\pi0$’s generalization to Background and New obj variants, and exhibits a safety property where it falls back to the base policy under unfamiliar conditions. [section A.3, Table 6]

### For reviewer-specific concerns, we additionally:
- Clarify the choice of $a_{better}$ (vxu1–W2) as a deliberate per-step proxy aligned with RoVer’s direction-focused contribution, and discuss sequence-modeling extensions as future work.
- Justify the loss weighting in Eq. (14) and show via ablations that direction supervision improves PRM learning even when direction-guided sampling is not used at inference (vxu1–Q2). [section A.6, Table 5, Figure 7]
- Provide a more detailed motivation for reallocating compute from training to inference, clarify the exact outputs of $R_\phi(h,a)$, explain the role of adaptive noise scale, and summarize our multi-task PRM training recipe and noise-scale sensitivity (nGo7–W1–W3). [section A.5, Table 4]
- Clarify PRM data reuse without extra annotation, and report additional latency measurements and direction-analysis discussion (xijL–W1–W4).

---
## General Author Statement

We also appreciate that the reviewers recognized several positive aspects of RoVer overall: the **novelty and cleverness** of the idea and PRM design (xijL), the **importance of the motivation and practical impact of the gains** in embodied AI (nGo7), RoVer’s **broad applicability** across different VLA backbones (UurV), and its **strong test-time scalability** in terms of performance improvements under additional compute (vxu1).

The main concerns are concentrated on (i) the comparison with V-GPS, (ii) compatibility with joint-angle action spaces, and (iii) validation on stronger VLA backbones such as **$\pi0$**. Our rebuttal adds targeted experiments and clarifications on all three aspects, and we believe these additions substantially address the reviewers’ questions.

We hope this summary helps you quickly connect the key concerns with the corresponding rebuttal updates and appended experiments.

Thank you for your time and consideration.

Sincerely, Authors

---

### Meta-Review · Area_Chair_vBrN · 2025-12-16

**Summary:**

The core reviewers' comments were mainly insufficient motivation; missing baseline; and vague method details. The reviewers pointed out that PRM still reuses the same dataset as the basic strategy, and it should be clarified that it is not "zero additional data"; the method only tests end-effector delta actions, and its generalization to joint angle or velocity control needs to be verified; the real robot is only paired with Diffusion Policy, lacking advanced VLA experimental comparisons; the orientation prediction shows limited improvement on PushT, and its failure/effectiveness should be analyzed.

**Reviewer Concerns:**

Although the authors have supplemented their responses with experiments, ablation, and baseline comparisons, the proposed method shows very limited performance improvement compared to the baseline, and its generalizability and effectiveness remain to be addressed.

**Reviewer Scores:**

The reviewers consistently maintained negative scores.

---

### Decision · Program_Chairs · 2026-01-26

Reject